# Modeling the drivers of fine PM pollution over Central Europe: impacts and contributions of emissions from different sources

Lukáš Bartík[1], Peter Huszár[1], Jan Karlický[1], Ondřej Vlček[2], and Kryštof Eben[3]

[1]Department of Atmospheric Physics, Faculty of Mathematics and Physics, Charles University, Prague, V Holešovičkách 2, 18000 Prague 8, Czech Republic
[2]Czech Hydrometeorological Institute, Na Šabatce 2050/17, 143 06 Prague 4, Czech Republic
[3]Czech Academy of Sciences, Institute of Computer Science (ICS), Pod Vodárenskou věží 271/2, 182 00 Prague 8, Czech Republic

**Correspondence:** Lukáš Bartík (lukas.bartik@matfyz.cuni.cz)

**Abstract.** Air pollution nowadays represents the most significant environmental health risk in Europe, with fine particulate matter ($PM_{2.5}$) being among the pollutants with the most critical threat to human health, especially in urban areas. Identifying and quantifying the sources of $PM_{2.5}$ components are essential prerequisites for designing effective strategies to mitigate this kind of air pollution. In this study, we utilized the Weather Research and Forecast Model (WRF) coupled with the Comprehensive Air Quality Model with Extensions (CAMx) to investigate the relationships between emissions and the concentrations of total $PM_{2.5}$ and its secondary components over Central Europe during the period 2018–2019, with a more detailed focus on six large cities in the region, namely Berlin, Munich, Vienna, Prague, Budapest, and Warsaw.

Concretely, we conducted three experiments named PSAT, SOAP, and VBS. In the PSAT experiment, we used the Particulate Source Apportionment Technology (PSAT) built into CAMx to determine the contributions of individual anthropogenic emissions sectors and total biogenic emissions to the mentioned concentrations. The SOAP and VBS experiments represent sensitivity analyses based on the zero-out method, using which we determined the impacts of the complete reduction of emissions from individual anthropogenic emissions sectors on the mentioned concentrations. The difference between the SOAP and VBS experiments lies in using different model mechanisms of secondary organic aerosol formation built into CAMx and associated emission estimates of intermediate-volatility and semivolatile organic compounds, which allowed us to evaluate the sensitivity of their use to the resulting impacts. While we used the Secondary Organic Aerosol Processor (SOAP) in the SOAP experiment, we employed the 1.5-dimensional volatility basis set (1.5-D VBS) in the VBS experiment. The overall design of the study, in which anthropogenic emissions were divided into 12 emission sectors defined by the Gridded Nomenclature For Reporting, makes it the only one of its kind for this region.

The use of the PSAT tool showed, among other things, that during the winter seasons, emissions from other stationary combustion (including residential combustion), boundary conditions, road transport, and agriculture–livestock contribute most extensively to the average $PM_{2.5}$ concentrations, with domain-wide average contributions of 3.2, 2.1, 1.4, and 0.9 $\mu g\,m^{-3}$, respectively, while during the summer seasons, the average $PM_{2.5}$ concentrations are mainly contributed by biogenic emissions, followed by emissions from road transport, industrial sources, and boundary conditions, with domain-wide average contributions of 0.57, 0.31, 0.28, and 0.27 $\mu g\,m^{-3}$, respectively. In contrast, the most considerable average seasonal impacts on $PM_{2.5}$

concentration in the SOAP experiment are caused by the overall reduction of emissions from other stationary combustion, agriculture–livestock, road transport, and agriculture–other during the winter seasons, with domain-wide averages of 3.4, 2.9, 1.4, and 1.1 $\mu g\,m^{-3}$, respectively, while during the summer seasons, they are induced by emissions from agriculture–livestock, road transport, industrial sources, and other stationary combustion, with domain-wide averages of 0.46, 0.45, 0.34, and 0.29 $\mu g\,m^{-3}$, respectively. Further, we revealed that the differences between the contributions of emissions from anthropogenic sectors to $PM_{2.5}$ concentration and the impacts of these emissions on $PM_{2.5}$ concentration in the SOAP experiment are predominantly caused by the acting of oxidation–limiting and/or indirect effects on the secondary aerosol components. Moreover, the most substantial of these differences, in terms of daily averages in the cities and seasonal averages for the winter and summer seasons over the domain, are associated with emissions from agriculture–livestock, mainly due to differences in nitrate concentrations. Specifically, in the case of the daily averages, they reached up to around 15 $\mu g\,m^{-3}$ in some cities during winter time, while in the case of the seasonal averages, they reached up to 4.5 and 1.25 $\mu g\,m^{-3}$, respectively. Finally, the comparison of the SOAP and VBS experiments showed that the modifications used in the VBS experiment mainly increase the average seasonal impacts on $PM_{2.5}$ concentration caused by the overall reduction of emissions from other stationary combustion and road transport during the winter seasons, while during the summer seasons, they do the same mainly for the overall reduction of emissions from road transport. These increases reached up to 12 and 4 $\mu g\,m^{-3}$, respectively, during the winter seasons and up to 2.25 $\mu g\,m^{-3}$ during the summer seasons.

## 1 Introduction

Particulate matter (PM) is a component of ambient air pollution that is widely recognized for its harmful effects on human health, including various respiratory and cardiovascular problems that can result in premature death (e.g., Anderson et al., 2012; Apte et al., 2015; Turner et al., 2020). According to the European Environment Agency's latest report on air quality in Europe (EEA, 2022), air pollution is the most significant environmental health risk in Europe, which significantly impacts the health of the European population, particularly in urban areas. Regarding PM with an aerodynamic diameter $\leq 2.5\,\mu m$ ($PM_{2.5}$, also called fine PM), the report concludes that in 2020, 96 % of the urban population in the European Union was exposed to levels above the health-based guideline level for it set by the World Health Organization (5 $\mu g\,m^{-3}$), which resulted in 238,000 premature deaths.

Although the chemical composition of fine PM (including submicron PM) in Central Europe shows significant spatial and temporal variability, it is generally dominated by organic matter and secondary inorganic aerosols (e.g., Lanz et al., 2010; Putaud et al., 2010; Szigeti et al., 2015; Schwarz et al., 2016; Juda-Rezler et al., 2020; Bressi et al., 2021; Chen et al., 2022). Moreover, Chen et al. (2022) suggested that secondary organic aerosol (SOA) is the main contributor to total submicron PM and dominates organic aerosol across Europe.

In order to design effective strategies to mitigate the adverse effects of PM, it is essential to thoroughly understand PM sources, which is still a challenge as PM consists of a host of components with different sources and atmospheric behavior (Hendriks et al., 2013). One of the commonly used ways to source attribution analysis of PM is to use sophisticated Eulerian

chemical transport models (CTMs) such as the Comprehensive Air Quality Model with Extensions (CAMx; Ramboll, 2022), the Community Multiscale Air Quality (CMAQ) model (EPA, 2022) or CHIMERE (LMD, 2022). It is given by the fact that these models can describe not only the evolution of primary PM but also contain modules that can rigorously control the formation of secondary inorganic and organic PM from gaseous precursors and its subsequent development, as well as aqueous aerosol chemistry.

Over time, several methods have been developed to study relationships between PM concentrations and emission sources using CTMs. Depending on the approach used for such an analysis of PM sources, they have been generally divided into sensitivity analysis methods and reactive tracer (also called tagged species) methods (e.g., Yarwood et al., 2007; Clappier et al., 2017). The fundamental difference between these two approaches lies in the following: while sensitivity analysis methods estimate the impact on pollutant concentration that results from a change of one or more emission sources, reactive tracers methods deal with a source apportionment, which means that they quantify the contribution of an emission source or precursor to the concentration of one pollutant at one given location (Clappier et al., 2017). It is also important to emphasize here that only in the case of linear (or close to linear) relationships between concentration and emissions, impacts given by sensitivity analysis methods and contributions given by reactive tracers methods are equivalent (or close) concepts (Clappier et al., 2017).

One of the traditional sensitivity analysis methods, frequently used for PM source attribution due to its simplicity and intuitive interpretation, is the zero-out method, which is an extreme case of the brute-force method. As the name suggests, this method quantifies the impact of a particular emission source by comparing the model outputs of a base simulation, in which emissions from all sources were taken into account, with the outputs of a perturbed simulation, in which emissions from the source of interest were set to zero, because it seems intuitively obvious that removing a source should reveal the source's impact (Yarwood et al., 2007). Using this method for experiments with many studied emission sources quickly becomes impractical and computationally demanding, as it requires the implementation of a large number of perturbed simulations. Among the works in which the zero-out method was used to study the impacts of anthropogenic activity sectors on the total concentrations of fine PM in various regions of Europe, we mention the papers of Tagaris et al. (2015), Jiménez-Guerrero (2022), and Arasa et al. (2016) as they differ from most other ones in that their authors used the zero-out method to determine impacts of either all or almost all of anthropogenic activity sectors within the SNAP (Standard Nomenclature for Air Pollution) classification. Concretely, Tagaris et al. (2015) studied these impacts over the whole of Europe but on a model domain with a relatively coarse horizontal resolution (35 km) and only for one month (July 2006). Jiménez-Guerrero (2022) did the same over the Iberian Peninsula using a model domain with a horizontal resolution of 9 km for the summer (June–August 2011) and winter (December 2011–February 2012) scenarios. Finally, Arasa et al. (2016) made such a sensitivity analysis for the region of Madrid and the urban metropolitan area of Madrid on model domains with a horizontal resolution of 3 km and 1 km, respectively, for the year 2010.

Unlike the zero-out method, which can be applied in any CTM, the selection of the tagged species method for PM source apportionment is limited by the selection of a CTM since usually only one such method, if any, is implemented in each CTM. For example, while the CAMx model provides the PSAT (Particulate Source Apportionment Technology; Yarwood et al., 2007; Ramboll, 2022) module for this purpose, the TSSA (Tagged Species Source Apportionment; Wang et al., 2009) module

can be used in older versions of the CMAQ model, and the ISAM (Integrated Source Apportionment Method; EPA, 2022) module in its newer versions. CAMx, like any other Eulerian CTM, naturally cannot provide any source apportionment in its

'normal' calculations, as it mixes all emissions from different sources together during them. In order to perform PM source apportionment within a CAMx simulation, the PSAT module employs sets of several families of reactive tracers, which are added for each emission source category/region to track the effects of emissions, transport, diffusion, deposition, chemical reactions, and initial and boundary conditions. Therefore, the very use of this tool requires having properly allocated emission sources, which can be defined in terms of geographical regions, emission categories or their groups, and initial and boundary

conditions. The significant flexibility of this module enables the implementation of a complex PM source apportionment, including several emission categories from several geographical regions in one model simulation; however, the increase in complexity also significantly affects computational demands.

Tagged species methods have been used in several studies dealing with the origin of fine PM in various regions of Europe. Hendriks et al. (2013) used the LOTOS-EUROS model (Schaap et al., 2008) equipped with a source apportionment module

based on the PSAT approach (Kranenburg et al., 2013) to establish the origin of ambient PM ($PM_{10}$ and $PM_{2.5}$) over the Netherlands for the years 2007–2009. Skyllakou et al. (2014) used the Particulate Matter Comprehensive Air Quality Model with Extensions (PMCAMx; Fountoukis et al., 2011) together with their extension of the PSAT algorithm (Wagstrom et al., 2008) over Europe on a model domain with a horizontal resolution of 36 km to estimate the impact of local emissions and pollutant transport on primary and secondary fine PM mass concentration levels in Paris during the summer of 2009 and the

winter of 2010. Bove et al. (2014) used CAMx version 5.2 combined with the PSAT module on model domains covering Europe and the area around the city of Genoa, Italy, with a horizontal resolution of 10 km and 1.1 km, respectively, to estimate major $PM_{2.5}$ emission sources in the city during a summer and late autumn period in 2011, which they subsequently compared with the estimates achieved from Positive Matrix Factorization. Karamchandani et al. (2017) used the PSAT method in CAMx version 6.1 on a model domain with a horizontal resolution of 23 km to identify the main source sectors of fine PM in 16 major

European cities, including Berlin, Germany; Warsaw, Poland; and Budapest, Hungary, from the Central European region, during February and August of 2010. Skyllakou et al. (2017) used PMCAMx combined with the extended PSAT algorithm of Skyllakou et al. (2014) over Europe on a model domain with a horizontal resolution of 36 km in order to quantify the sources that contribute to the primary and secondary organic aerosol during three different periods in 2008 and 2009. Pepe et al. (2019) used CAMx version 6.3 together with the PSAT module on model domains covering the Po Valley and the metropolitan area of

Milan with a horizontal resolution of 5 km and 1.7 km, respectively, to perform multi-pollutant source apportionment analyses, including $PM_{2.5}$, that combine emission categories and regions for the calendar year of 2010. Coelho et al. (2022) used CAMx version 6.3 together with the PSAT tool to, among other things, quantify the main sources of $PM_{2.5}$ and $PM_{10}$ over four European urban areas, including Sosnowiec, Poland, from the Central European region, for the year 2010. Finally, Pültz et al. (2023) used the LOTOS-EUROS model version 2.1 together with the PSAT algorithm on a European domain with a horizontal

resolution of about $28 \times 32$ $km^2$ with a nested domain covering Germany, Poland, and the Czech Republic with a horizontal resolution of about $7 \times 8$ $km^2$ to identify the most relevant sources of PM in the Berlin agglomeration area, Germany, covering the period from 2016 to 2018.

In this work, we use an offline coupled modeling framework consisting of a numerical weather prediction model and a CTM on the Central European domain with a moderate horizontal resolution (9 km) to perform: (1) two sensitivity analyses quantifying the impacts of emissions from a wide range of anthropogenic activity sectors on the concentrations of PM$_{2.5}$ and its secondary components (ammonium, nitrate, sulfate, and secondary organic aerosol) using the zero-out method, and (2) source apportionment to estimate the contributions of emissions from the same sectors of anthropogenic activity used in the sensitivity analyses to the concentrations of PM$_{2.5}$ and its secondary components using the PSAT tool, both for the relatively current period covering the years 2018 and 2019. Moreover, in addition to analyzing the outputs determined using both methods over the entire Central European domain, we also focus on six large cities in this region: Prague, Czechia; Berlin, Germany; Munich, Germany; Vienna, Austria; Budapest, Hungary; and Warsaw, Poland. Compared to the previous works mentioned above, ours is exceptional in that it is the first to implement both approaches, i.e., sensitivity analysis and source apportionment, simultaneously in one of the regions of Europe.

## 2    Methodology

### 2.1    Models and their configurations used

To describe the regional weather conditions and to drive the chemistry transport model, the Weather Research and Forecast (WRF) Model version 4.2 was adopted in our study. To simulate the chemistry and transport of pollutants, CAMx version 7.10 was used.

The WRF is an atmospheric modeling system designed for research and numerical weather prediction whose detailed description can be found in Skamarock et al. (2019). Our setup handled long- and short-wave radiation transfer using the Rapid Radiative Transfer Model for General Circulation Models (RRTMG; Iacono et al., 2008). Land-surface processes were driven using the Noah land-surface model (Chen and Dudhia, 2001). Urban canopy meteorological effects were invoked by a bulk approach, which treats urban surfaces as any other flat surfaces with physical parameters specific to urban surfaces (like roughness, albedo, etc.). Microphysical processes were parameterized using the scheme proposed by Thompson et al. (2008). Turbulent exchange in the planetary boundary layer (PBL) was solved by the BouLac PBL scheme (Bougeault and Lacarrere, 1989), and convection was calculated using the modified version of the Kain-Fritsch scheme (Kain, 2004).

The CAMx is a state-of-the-science Eulerian chemical transport model, a detailed description of which can be found in Ramboll (2022). To solve the gas-phase chemistry, we applied the CB6r5 mechanism (5th revision of the Carbon Bond mechanism version 6), developed initially as the CB6 by Yarwood et al. (2010), and since then, several times revised. The CB6r5 mechanism consists of 233 reactions among 87 species (62 state gases and 25 radicals) that can also be found in Ramboll (2022). The mechanism was numerically solved using an implementation of the Euler Backward Iterative (EBI) method developed by Hertel et al. (1993).

We used a static two-mode coarse/fine (CF) scheme to run aerosol chemistry processes together with the gas-phase chemistry. In this scheme, which divides the aerosol size distribution into two static modes (coarse and fine), primary species can be modeled as fine and/or coarse particles. In our case, both modes were considered. In contrast, all secondary species are

modeled as fine particles only. Aqueous aerosol formation in resolved cloud water was driven using the modified version of the RADM (Regional Acid Deposition Model) aqueous chemistry algorithm (Ramboll, 2022), developed initially by Chang et al. (1987). To predict the physical state and composition of inorganic aerosols, we applied the thermodynamic equilibrium model ISORROPIA version 1.7 (Nenes et al., 1998, 1999), which solves partitioning between the gas and aerosol phases for the sodium–ammonium–chloride–sulfate–nitrate–water aerosol system, with an update for calcium nitrate on dust particles.

Two modules can solve organic aerosol-gas partitioning and oxidation chemistry in CAMx version 7.10, and we applied both in the sensitivity analyses, as will be mentioned in more detail later. The first one is the Secondary Organic Aerosol Processor (SOAP) version 2.2, developed initially by Strader et al. (1999) and subsequently updated over time. The description of its recent version can be found in Ramboll (2022). Shortly, this module: (1) treats primary organic aerosol (POA) as a single non-volatile species that does not chemically evolve, and (2) considers oxidation of seven gaseous precursors belonging to anthropogenic and biogenic VOCs (volatile organic compounds) to form three semi-volatile surrogate compounds for each VOC precursor that can coexist in the gas and aerosol phases based on the pseudo-ideal solution theory of Odum et al. (1996). The second module, the 1.5-dimensional (1.5-D) volatility basis set (VBS), represents a hybrid VBS approach that provides a unified framework for gas-aerosol partitioning and chemical aging of both primary and secondary organic aerosol (Koo et al., 2014). It combines the simplicity of the one-dimensional VBS approach proposed by Donahue et al. (2006), in which the evolution of organic aerosol (OA) is described using a set of semi-volatile OA species with volatility equally spaced in a logarithmic scale (the basis set), with the ability to describe the OA evolution in the two-dimensional (2-D) space of oxidation state and volatility used in the (2-D) VBS approach (Donahue et al., 2011, 2012) by using multiple reaction trajectories defined in the 2-D VBS space. Namely, the 1.5-D VBS scheme uses five basis sets to describe varying degrees of oxidation in ambient OA: three for freshly emitted OA (hydrocarbon-like OA from meat cooking and other anthropogenic sources and biomass burning OA) and two for chemically aged oxygenated OA (anthropogenic and biogenic).

As we mentioned in the introduction, for PM source apportionment in CAMx, it is possible to use the PSAT tool, proposed initially by Yarwood et al. (2007). The PSAT modification implemented in the CAMx version we used, a detailed description of which can be found in Ramboll (2022), enables source apportionment of primary PM, ammonium ($PNH_4$), nitrate ($PNO_3$), sulfate ($PSO_4$), SOA, and particulate mercury using a total of 42 tracers for each source region/group. The flexibility of this implementation makes it possible to reduce the number of considered PM species and, thus, also the necessary tracers. In our case, we did not consider the source apportionment of particulate mercury and eight primary elemental species (e.g., iron, manganese, or silicon), which can be included by invoking the extended version of the CF scheme. One of the drawbacks of the current implementation of the PSAT tool in the model is that it only describes the OA mass based on the SOAP approach.

To solve the dry deposition of gases and aerosols, we used the methods of Zhang et al. (2003) and Zhang et al. (2001), respectively. Finally, to calculate the wet deposition of gases and aerosols, we applied the CAMx wet deposition model, a detailed description of which can be found in Ramboll (2022). The model employs a scavenging approach in which scavenging coefficients are determined on the relationships described by Seinfeld and Pandis (1998).

## 2.2 Model domains and input data

As mentioned in the introduction, we used an offline coupled model framework of the models described above (i.e., without assuming feedback of air pollutants to processes governing weather conditions) to achieve the goals of this paper. In other words, we first performed a regional weather simulation using the WRF model, the outputs of which we subsequently used to create the required meteorological input fields for all CAMx simulations performed.

The regional weather simulation was conducted on the Central European model domain centered over Prague (50.075° N, 14.44° E), Czechia, with a horizontal resolution of 9 km × 9 km that (1) contained 208 × 208 × 49 grid boxes in $x$, $y$, and $z$ directions, respectively, (2) reached the isobaric level of 50 hPa while the lowermost layer was about 48–50 m thick, and (3) used the Lambert conformal conic map projection. To force this simulation, we used the ERA-interim reanalysis (Simmons et al., 2010). All CAMx simulations were run on one domain, which had the same centering, horizontal resolution, and map projection as the WRF domain but was somewhat smaller compared to it. Concretely, it consisted of 172 × 152 × 20 grid boxes, with the vertical structure identical to the lowest 20 WRF domain layers and reaching approximately 12 km. The model orography of this domain and the locations of the analyzed cities are presented in Fig. 1. To create the required meteorological fields for CAMx simulations from the outputs of the weather simulation, we used the WRFCAMx preprocessor. This preprocessor is supplied with the CAMx code (https://www.camx.com/download/support-software/). One of the key parameters the WRFCAMx preprocessor calculates is the vertical eddy-diffusion coefficient that is shown to be the dominant driver of urban air pollution (Huszar et al., 2020b, a). In this study, the CMAQ method (Byun and Ching, 1999) was applied for its calculation.

Regarding anthropogenic emissions, we used three different emission inventories: (1) For the areas on the CAMx domain outside the Czech Republic, we applied the emissions from the CAMS (Copernicus Atmosphere Monitoring Service) European anthropogenic emissions - Air Pollutants inventory version 4.2 (Kuenen et al., 2021) for the year 2018. (2) For the area on the domain covering the Czech Republic, we adopted the high-resolution emissions from the Register of Emissions and Air Pollution Sources (REZZO – Registr emisí a zdrojů znečištění ovzduší) for the year 2018 issued by the Czech Hydrometeorological Institute (https://www.chmi.cz) together with the emissions from the ATEM Traffic Emissions dataset for the year 2016 provided by ATEM (Ateliér ekologických modelů – Studio of Ecological Models; https://www.atem.cz). These inventories provide annual emission totals of carbon monoxide (CO), sulfur dioxide ($SO_2$), nitrogen oxides ($NO_x$), ammonia ($NH_3$), methane ($CH_4$), non-methane VOCs (NMVOCs), and particulate matter aggregated to 12 GNFR (Gridded Nomenclature For Reporting) sectors of anthropogenic activity that are summarized in Table 1. To prepare the data from the mentioned emission inventories to emission files readable by CAMx, including preprocessing of the raw input files, the spatial redistribution of the annual emission totals into the grid of the CAMx domain, chemical speciation, and time disaggregation from annual to hourly emissions, we used the FUME (Flexible Universal Processor for Modeling Emissions) emission model (http://fume-ep.org/; Benešová et al., 2018). For chemical speciation, we used the speciation factors from Passant (2002). For time disaggregation, we applied sector-specific time disaggregation profiles proposed by Denier van der Gon et al. (2011).

Emissions of biogenic volatile organic compounds (BVOCs) were calculated using the Model of Emissions of Gases and Aerosols from Nature (MEGAN) version 2.1 (Guenther et al., 2012) driven by the weather conditions obtained from the

regional weather simulation. Vegetation characteristics needed for this model simulation, e.i., plant functional types, emission factors, and leaf-area-index data, were derived based on Sindelarova et al. (2014).

### 2.2.1 Estimates of I/SVOCs emissions

Because emissions of intermediate-volatility organic compounds (IVOCs) and semivolatile organic compounds (SVOCs), which are considered to be important precursors of SOA, are generally missing in current emission inventories, it is common for CTM modeling purposes to estimate them in the form of surrogate species based on sector-specific (alternatively on non-sector-specific) parametrizations (e.g., Giani et al., 2019; Jiang et al., 2019b, 2021). With the intention of including these emissions in our model experiments, we proceeded analogously.

Specifically, to estimate IVOCs and SVOCs emissions produced by gasoline and diesel vehicles, we adopted the methodology used by Giani et al. (2019). Thus, we first estimated IVOCs emissions for gasoline and diesel vehicles as 0.0397 and 1.2748 times their corresponding NMVOCs emissions, respectively. Next, we estimated emissions of organic matter in the semivolatile range ($OM_{SV}$) based on the estimates of IVOCs emissions and using knowledge of the ratio of IVOCs emissions to $OM_{SV}$ emissions, $R$ ($R = 4.62$ for gasoline vehicles and $R = 2.54$ for diesel vehicles), derived from the volatility distribution for gasoline and diesel vehicles provided by Zhao et al. (2015) and Zhao et al. (2016), respectively. Furthermore, we used these distributions to redistribute $OM_{SV}$ of both sources into the volatility bins used in the 1.5-D VBS scheme.

Following the methodology justified by Ciarelli et al. (2017) and also used by Jiang et al. (2019b, 2021), we estimated IVOCs emissions from biomass burning as 4.5 times POA emissions summed up from other stationary combustion and agriculture–other. In the territory of the Czech Republic, where we used more detailed data on residential combustion, we applied this parametrization only to the part of POA produced by wood combustion. The IVOCs emissions from other anthropogenic sources we calculated as 1.5 times their corresponding POA emissions, as Robinson et al. (2007) proposed. Finally, to offset the influence of missing SVOCs emissions from biomass burning and other anthropogenic sources besides gasoline and diesel vehicles, we adopted the routinely used approach of multiplying their corresponding POA emissions by a factor of 3 (e.g., Jiang et al., 2019b, 2021).

For the sake of completeness, we add that we considered only IVOCs estimates in all CAMx simulations using the SOAP module since POA is regarded as non-volatile in this case, while in those implemented using the 1.5-D VBS module, we naturally considered both IVOCs and SVOCs estimates.

### 2.3 Model experiments: design, validation, and evaluation

Because our main objective is to assess the impacts and contributions of emissions from the broadest possible range of anthropogenic activity on fine PM and its secondary components, and we use the emission inventories that classify anthropogenic activity into 12 GNFR sectors A–L, we decided to design model experiments so that they evaluate the impacts and contributions of all 12 GNFR sectors separately. Another aspect we considered is the dual implementation of the organic aerosol chemistry/partitioning using either the SOAP module or the 1.5-D VBS module. Hence, to assess the influence of these different implementations on the sector impacts, we conducted two sensitivity experiments based on the zero-out method, each

using one of the modules in all of its CAMx simulations. We further label them as the SOAP and VBS experiments based on the module employed. In order to meet the mentioned experimental design, both of these sensitivity experiments consist of one base simulation, in which the total emissions from all sources (i.e., anthropogenic and biogenic sources and boundary conditions) were considered, and 12 perturbed simulations, in which emissions from one GNFR sector (different in each of these simulations) were removed from the total emissions.

As the applicability of the PSAT tool is conditioned by utilizing the SOAP module during a CAMx simulation, we performed only one experiment to determine the PM source apportionment using this tool. This experiment, further labeled as the PSAT experiment, evaluates the contributions of the individual GNFR sectors, biogenic emissions, and initial and boundary conditions in one simulation, thanks to the flexibility of the PSAT tool mentioned in the introduction. To achieve this, we have prepared the emission inputs divided into the relevant categories (i.e., into the individual GNFR sectors, biogenic emissions, and boundary conditions) for this simulation. The different approach in providing emissions (total vs. categorized emissions) is the only difference in the model setup between the base simulation of the SOAP experiment and the simulation of the PSAT experiment. Hence, for each chemical species, the sum of all contributions to its concentration in the PSAT experiment should correspond to its concentration in the base simulation of the SOAP experiment. The basic parameters of all three mentioned experiments are summarized in Table 2.

To demonstrate the capabilities and shortcomings of the model system we used, we validated the modeled concentrations of $PM_{2.5}$ and some of its components and gaseous precursors. Specifically, in the case of $PM_{2.5}$ components, we focused on $PNH_4$, $PNO_3$, $PSO_4$, elemental carbon (EC), and organic carbon (OC), while in the case of gaseous precursors, we focused on nitrogen dioxide ($NO_2$) and $SO_2$. Naturally, we used only the simulation of the PSAT experiment and the base simulations of the SOAP and VBS experiments to validate the modeled concentrations because, by the nature of their construction, only these three are different model representations of reality. At the same time, taking into account the horizontal resolution used in all these simulations (9 km), we considered it reasonable to compare them only with the measurements at the background stations located up to 800 meters above sea level, which additionally covered at least 75 % of the modeled period. For $PM_{2.5}$, $NO_2$, and $SO_2$, we selected such measurements at Czech, German, Austrian, Hungarian, Polish, and Slovak rural, suburban, and urban background stations from the AirBase database provided by the European Environmental Agency (https://discomap.eea.europa.eu/map/fme/AirQualityExport.htm). The list of all these stations is given in Table S1, provided in the Supplement. For the $PM_{2.5}$ components, whose systematic long-term monitoring in the Central European region is considerably spatially limited and concentrated in rural areas, we selected their measurements at the suitable rural background stations included in the Cooperative Programme for Monitoring and Evaluation of the Long-range Transmission of Air Pollutants in Europe (EMEP), as well as at one suitable rural background station not included in the EMEP. The list of all these stations is provided in Table S2. As can be seen in this table, some of the stations were taken from the EBAS database (https://ebas-data.nilu.no/default.aspx), whereas the rest were taken from the AirBase database.

As part of the validation process, we first compared the measured and modeled $PM_{2.5}$ daily concentrations in the selected cities during the winter (December–January–February), spring (March–April–May), summer (June–July–August), and autumn (September–October–November) seasons of 2018–2019 using Pearson correlation coefficient ($r$), normalized mean bias

(NMB), and normalized mean square error (NMSE), the definitions of which are given by Eq. (S1)–(S3) in the Supplement. Specifically, we analyzed the seasonal values of these statistical indicators averaged over all suitable urban and suburban background stations in the selected cities, the list of which is summarized in Table S3. Further, we compared the measured and modeled annual cycles of the monthly concentrations of the mentioned pollutants averaged over suitable stations. Specifically, for $PM_{2.5}$, $NO_2$, and $SO_2$, we first carried out such comparisons at the level of the individual studied cities, using the urban and suburban background stations listed in Table S3. Subsequently, we also performed them for all the rural background stations and all the suburban and urban background stations listed in Table S1. Finally, for $PNH_4$, $PNO_3$, $PSO_4$, EC, and OC, we made analogous comparisons using the rural background stations listed in Table S2.

Since meteorological conditions influence the concentrations of $PM_{2.5}$ and its components, it is also appropriate to validate how well the WRF model represents such conditions in our simulation. To get at least a partial idea of this in a specific part of the domain, we compared the measured and modeled hourly values of both air temperature measured at 2 m above the ground and wind speed measured at 10 m above the ground at all Prague synoptic stations listed in Table S4. Specifically, we first compared the annual cycles of their monthly means averaged over all the stations and then the diurnal cycles of their seasonal means averaged over all the stations in the winter and summer seasons. The relevant measurements of air temperature and wind speed were provided to us by the Czech Hydrometeorological Institute (https://www.chmi.cz).

When evaluating the impacts and contributions, we focused on their average temporal absolute/relative impacts and contributions, the definitions of which are given in Appendix A. More precisely, when assessing the spatial distributions of the impacts and contributions over Central Europe and its surrounding areas, we focused on the average seasonal absolute/relative impacts and contributions, specifically for the winter and summer seasons. In order to provide information about the contributions and impacts of emissions even at a greater temporal resolution, in the case of their evaluation in the selected cities, we focused on the average daily absolute/relative impacts and contributions. In addition, we also determined their seasonal averages in the winter and summer seasons. Before the evaluation, we removed the first 14 days (1–14 January 2018) from all the simulations, viewing them as a spin-up time. We also did the same before validating the simulations.

## 3  Results

### 3.1  Validation

Table 3 shows the average statistical indicators ($r$, NMB, and NMSE) comparing the modeled and measured daily $PM_{2.5}$ concentrations during the individual seasons in all the studied cities. Regarding the correlations, the modeled concentrations in all three simulations correlate best with the measurements during the winter seasons ($r$ = 0.66–0.82) in all the studied cities except Vienna, where it occurs in the spring seasons ($r$ = 0.73–0.74). On the contrary, the worst correlated in all three simulations are almost exclusively the concentrations in the summer seasons ($r$ = 0.28–0.55). The average NMB values indicate that the modeled concentrations in all three simulations, excluding those in Prague and Munich during the winter seasons, are, on average, underestimated compared to the measurements. The greatest underestimations are observed during the summer seasons, with an average NMB from -75.8 to -35.1 %. In contrast, the smallest deviations between the modeled and measured

concentrations, in terms of the absolute value of the average NMB, are most common in the winter seasons. The best agreements
with the measurements, where the average NMB does not exceed 10 %, are achieved in several cases. These include the base
simulation of the VBS experiment in Munich during the autumn seasons (-0.4 %) and Budapest during the winter seasons
(-3.0 %), the base simulation of the SOAP experiment in Munich and Prague during the winter seasons (1.5 % and 5.4 %,
respectively), and the simulation of the PSAT experiment in Munich and Prague during the winter seasons (1.6 % and 5.4 %,
respectively). The average NMSEs for all three simulations in all the cities are almost always the smallest (NMSE = 21.9–49.7
%) during the winter periods. On the contrary, they are almost always the largest during the summer periods (NMSE = 39.5–
274.3 %). At the same time, the average NMSE values for the base simulation of the VBS experiment are almost always more
or less smaller than those for the other two simulations. Finally, it is essential to point out the striking similarity of all three
indicators for the base simulation of the SOAP experiment with those for the simulation of the PSAT experiment in all the
cities during all the seasons, which shows and partially proves the expected high consistency of the model in the prediction of
individual PM components during the simulation with and without the use of the PSAT tool.

Figure 2 compares the average modeled and measured annual cycles of average monthly $PM_{2.5}$ concentrations in all the
studied cities. As regards the modeled monthly averages, it is seen that those in the PSAT experiment are almost identical
to those in the base simulation of the SOAP experiment in all the cities during all months, which again points to the above-
mentioned high consistency of the model. At the same time, the modeled monthly averages in both of these simulations are
always smaller than their corresponding monthly averages in the base simulation of the VBS experiment: the differences
between them are most often up to 2 $\mu g\,m^{-3}$. The comparison further reveals a certain spatiotemporal conditionality of the
model's ability to predict the monthly averages. In Berlin, Vienna, and Warsaw, the model underestimates them all year round
in all three cases. In Budapest and Prague, the model fails in the same way in capturing the monthly averages during the warm
half-year (April–September) and other autumn months in all the cases; however, it captures them relatively accurately in most
of the remaining months. Finally, in Munich, the model underestimates the monthly averages in all three cases from March to
August but sets them excellently during all autumn months in the base simulation of the VBS experiment and during all winter
in the base simulation of the SOAP experiment.

The average modeled and measured annual cycles of average monthly $NO_2$ and $SO_2$ concentrations in the individual cities
are depicted in Fig. S1. The average modeled cycles for $SO_2$ are identical in all three simulations, while those for $NO_2$ are
almost identical, with slight differences occurring in the warm half of the years. As for $NO_2$, the model can capture the shape
of the average measured cycle relatively well in all the cities, but it always more or less underestimates it, usually by about
8–20 $\mu g\,m^{-3}$. On the other hand, the ability of the model to capture the average measured cycle for $SO_2$ varies considerably in
the individual cities. In Vienna, the model captures them relatively well, with some exceptions. In Budapest, the model mainly
underestimates them, while in Warsaw, it usually enormously overestimates them. Further, Fig. S2 shows the average modeled
and measured annual cycles of average monthly $PM_{2.5}$, $NO_2$, and $SO_2$ concentrations over the rural stations, as well as over the
suburban and urban stations. Briefly, the average cycles for $PM_{2.5}$ show qualitatively similar behavior in both cases to the one
described above for Berlin, Vienna, and Warsaw. For $NO_2$, the average cycles in both cases qualitatively behave as described
above for the individual cities. As for $SO_2$, the model can capture quite well the average measured cycle over all the suburban

and urban stations in all three simulations, except for a few months. At the same time, over all the rural stations, the model captures it relatively accurately in the winter months and underestimates it by up to about 1.75 $\mu g\,m^{-3}$ in the other months.

Figure 3 illustrates the average modeled and measured annual cycles of average monthly $PNH_4$, $PNO_3$, $PSO_4$, EC, and OC concentrations. Except for OC, the modeled cycles for the other components are almost the same in all three simulations. The modeled average monthly OC concentrations in the base simulation of the VBS experiment are higher than their corresponding concentrations in the other two simulations during the whole year, with a maximum difference of up to 0.75 $\mu g\,m^{-3}$ in the winter months. Qualitatively, the model predicts the concentrations of the components in roughly two ways in all three simulations. First, for $PNH_4$, $PNO_4$, and EC, it overestimates them, with exceptions, from November to March, while in the remaining months, it tends to either underestimate them or determine them relatively accurately. Second, for $PSO_4$ and OC, it underestimates them throughout the year. The largest mentioned overestimations, reaching up to 2.5 $\mu g\,m^{-3}$, are associated with $PNO_3$. The most largely underestimated is the average monthly $PSO_4$, with values up to approximately 2.5 $\mu g\,m^{-3}$, and especially the average monthly OC, with values up to 4 $\mu g\,m^{-3}$, depending on the simulation being considered.

Finally, Fig. 4 presents a comparison between the average annual cycles of average monthly air temperatures and wind speeds during 2018-2019 in Prague, both modeled and measured, as well as the diurnal cycles of average seasonal air temperatures and wind speeds during the winter and summer seasons of the same period. Regarding the air temperatures, the WRF model accurately captures their average annual cycle, with values not exceeding 0.8 °C. As can be deduced from the average diurnal cycle for the winter seasons, the slightly higher average monthly air temperatures in the winter months are mainly caused by the overestimations of the air temperature at noon and in afternoon hours, whose seasonal average values reach up to 0.8 °C. Based on a similar argument for the summer seasons, the slightly lower average monthly air temperatures in the summer months are induced mainly by the underestimations of the air temperature in night hours, whose seasonal average values reach up to 2.3 °C. As for the wind speeds, the WRF model overestimates their average monthly values except for the summer months, whereby these overestimations reach up to about 0.9 $m\,s^{-1}$ in the winter months. The model overestimates their average diurnal cycle during the whole day in the winter seasons by 0.2–0.9 $m\,s^{-1}$. In the summer seasons, the model overestimates the averaged average seasonal wind speeds wind speeds by 0.1–0.2 $m\,s^{-1}$ in the evening and night hours, while in the rest of the day, it underestimates them by 0.1–0.8 $m\,s^{-1}$.

## 3.2 Spatial distributions of seasonal PM$_{2.5}$

Before describing the impacts and contributions during the winter and summer seasons, we consider it appropriate to describe the spatial distributions of the modeled seasonal concentrations of $PM_{2.5}$ in the base simulations of the SOAP and VBS experiment during the respective seasons. Because the corresponding distributions in the base simulation of the PSAT experiment are almost identical to those in the base simulation of the SOAP experiment, it is not necessary to describe them explicitly.

Figure 5 depicts the above distributions in both base simulations and the difference (VBS - SOAP) between them. In both simulations, the average seasonal $PM_{2.5}$ concentrations in the winter seasons are consistently higher than those in the summer seasons, except for several areas in the Alps. The domain average of their ratio (winter to summer) is 4.2 when using the SOAP scheme and 3.7 when using the VBS scheme.

In the base simulation of the SOAP experiment, the average concentrations during the winter seasons range from 1 to 35 $\mu g\,m^{-3}$ (Fig. 5a). The lowest values, reaching up to 3 $\mu g\,m^{-3}$, occur in the highest areas of the Alps. On the other hand, the Po Valley in Italy, most of Czechia (especially lowland and highly urbanized areas), some areas in southern and central Poland, some areas in the northern, southern, and central parts of the Pannonian Basin, and the central Slovenia area are the regions with the most pronounced $PM_{2.5}$ pollution. On most of the territory of the Po Valley, the average concentrations exceed 20 $\mu g\,m^{-3}$, and they exceed 14 $\mu g\,m^{-3}$ in other regions mentioned above. The distribution of the average seasonal $PM_{2.5}$ concentrations during the winter seasons in the base simulation of the VBS experiment (Fig. 5c), which range from 1 to 45 , is similar in its main features to that in the base simulation of the SOAP experiment. However, these two distributions differ quantitatively in that the seasonal concentrations in the base simulation of the VBS experiment are higher in all domain areas (Fig. 5e). Furthermore, these differences generally increase when approaching the regions corresponding to the most polluted regions in the base simulation of the SOAP experiment, reaching up to 14 $\mu g\,m^{-3}$ in the Po Valley.

During the summer seasons, the average seasonal $PM_{2.5}$ concentrations in the base simulation of the SOAP experiment reach up to 8 $\mu g\,m^{-3}$ but mostly do not exceed 3 $\mu g\,m^{-3}$ (Fig. 5b). The lowest values, reaching up to 1 $\mu g\,m^{-3}$, occur in the Alps and the central region of Slovakia. In contrast, higher values, ranging from 4 to 8 $\mu g\,m^{-3}$, are observed mainly in the Po Valley, the southern area of the Pannonian Basin, Silesia, Prague, and the southern and western regions of Germany. The corresponding average seasonal concentrations in the base simulation of the VBS experiment reach up to 10 $\mu g\,m^{-3}$ but mostly do not exceed 4 $\mu g\,m^{-3}$ (Fig. 5d). Compared to the average seasonal concentrations in the base simulation of the SOAP experiment, they are, analogously to the winter seasons, higher in all domain areas (Fig. 5f). The most pronounced differences between them, exceeding 1 $\mu g\,m^{-3}$, occur in the regions of the Po Valley, Silesia, and southern, central, and western Germany.

## 3.3 Spatial distributions of impacts and contributions

The following section highlights the most important results pertaining to the spatial distributions of the average seasonal impacts of emissions on $PM_{2.5}$ concentration in both sensitivity experiments. It also includes information on the spatial distributions of the average seasonal contributions of emissions to $PM_{2.5}$ concentration in the PSAT experiment and presents the main differences that arise from using both studied concepts. Additionally, it provides a similar analysis for $PNH_4$, $PNO_3$, $PSO_4$, and SOA.

### 3.3.1 $PM_{2.5}$

Figure 6 depicts the spatial distributions of the average seasonal absolute impacts of emissions from individual GNFR sectors on $PM_{2.5}$ concentration during the winter and summer seasons in the SOAP experiment. The corresponding spatial distributions of their average seasonal relative impacts are captured in Fig. S3. During the winter seasons (Figs. 6a and S3a), emissions from other stationary combustion, agriculture–livestock, road transport, agriculture–other, and industrial sources have the highest domain-wide absolute seasonal impacts on $PM_{2.5}$ concentration, reaching values of 3.4, 2.9, 1.4, 1.1, and 0.6 $\mu g\,m^{-3}$, respectively. Emissions from other stationary combustion have the most significant average seasonal absolute impacts in the areas with the most pronounced $PM_{2.5}$ pollution. In such areas, these impacts mostly exceed 6 $\mu g\,m^{-3}$ and reach up to 18 $\mu g\,m^{-3}$ in

some localities of the Po Valley, representing 40–60 % of the average seasonal PM$_{2.5}$ concentration. In other areas, they range between 1–6 $\mu g\, m^{-3}$, except for the highest areas of the Alps, where they are generally below 1 $\mu g\, m^{-3}$. The areas with these impacts between 4–6 $\mu g\, m^{-3}$ are mainly located in the peripheral areas of the Pannonian Basin and most of the territory of Poland. Emissions from agriculture–livestock give rise to the average seasonal absolute impacts of 2–4 $\mu g\, m^{-3}$ in most parts

of the domain, except for the Po Valley and central Poland area, where these impacts can go up to 8 and 6 $\mu g\, m^{-3}$, respectively. On the contrary, they are relatively lower in the Alps and central Slovakia region, with a maximum of 2 $\mu g\, m^{-3}$. Overall, the average seasonal absolute impacts of emissions from this sector dominate most of the territory of Germany, Switzerland, and the mountain areas of Austria, representing 25–50 % of the seasonal PM$_{2.5}$ concentration in these areas. Except for higher-lying areas of the domain, the average seasonal absolute impacts caused by emissions from road transport range between 1–6

$\mu g\, m^{-3}$, with values between 4–6 $\mu g\, m^{-3}$ being reached only in the Po Valley's central area and Prague. The corresponding average seasonal relative impacts lie mostly between 10–25 %, with higher values occurring especially in the western half of the domain. The last two sectors whose emissions cause the average seasonal absolute impacts higher than 1 $\mu g\, m^{-3}$, at least in specific domain locations, are industrial sources and shipping. The average seasonal absolute impacts caused by emissions from other sectors, including shipping for most of the domain, are either small (mostly up to 0.5 $\mu g\, m^{-3}$) or negligible over

most of the domain.

During the summer seasons (Figs. 6b and S3b), emissions from agriculture–livestock, road transport, industrial sources, other stationary combustion, and shipping have the highest domain-wide absolute seasonal impacts on PM$_{2.5}$ concentration, reaching values of 0.46, 0.45, 0.34, 0.29, and 0.20 $\mu g\, m^{-3}$, respectively. Moreover, these are the only anthropogenic emissions whose average seasonal impacts in the summer seasons exceed 0.5 $\mu g\, m^{-3}$ in larger areas of the domain and are even higher

than 1.5 $\mu g\, m^{-3}$ in its specific smaller locations. The location of these areas is strongly dependent on the emission sector. In the case of agriculture–livestock, these areas occur in most of the territory of Germany, some alpine localities of Switzerland and Austria, the areas of the Po Valley, and in the areas of central and eastern Poland, but the average seasonal absolute impacts range between 1–2 $\mu g\, m^{-3}$ only in northwestern Germany and the central area of the Po Valley. In connection with road transport, they are located in the Po Valley and on a vast area covering almost all of Germany, northern areas of Switzerland

and Austria, western Slovakia, the Czechia, and the southern and central regions of Poland, but the average seasonal absolute impacts range between 1–2 $\mu g\, m^{-3}$ only in the central area of the Po Valley, the regions of southern Germany, the regions of Czechia with high road traffic, expect Prague and its surroundings where they reach 1.5–2.5 $\mu g\, m^{-3}$. In the case of industrial sources, these areas occur mainly in western, southern, and eastern Germany, the Po Valley, central and southern Poland, eastern Bohemia, and Serbia. Moreover, in some regions of southern Poland and Serbia, their average seasonal absolute impacts reach

2–3 $\mu g\, m^{-3}$, representing the highest average seasonal absolute impacts during the summer seasons in the SOAP experiment. Concerning other stationary combustion, these areas are located in the Pannonian Basin and the Po Valley, but only in the central areas of the Po Valley, the average seasonal absolute impacts range between 1–2 $\mu g\, m^{-3}$. With regard to shipping, they are located in the Gulf of Venice and the southern and northwestern regions of Germany, but only in the coastal areas of northwestern Germany, the average seasonal absolute impacts range between 1–2 $\mu g\, m^{-3}$. Finally, the average seasonal

absolute impacts caused by emissions from other sectors are either negligible over most of Central Europe or range over it mostly between 0.05–0.5 $\mu g\,m^{-3}$.

The spatial distributions of the average seasonal absolute impacts of emissions from individual GNFR sectors on $PM_{2.5}$ concentration during the winter and summer seasons in the VBS experiment are shown in Fig. S4, while the corresponding spatial distributions of their average seasonal relative impacts are depicted in Fig. S5. As in the SOAP experiment, the sectors with the highest domain-wide average of the average seasonal absolute impacts during the winter seasons in this experiment are other stationary combustion (4.2 $\mu g\,m^{-3}$), agriculture–livestock (2.9 $\mu g\,m^{-3}$), road transport (1.7 $\mu g\,m^{-3}$), agriculture–other (1.1 $\mu g\,m^{-3}$), and industrial sources (0.6 $\mu g\,m^{-3}$). Here, in Fig. 7, we present the spatial distributions of the differences between the average seasonal absolute impacts on $PM_{2.5}$ concentration in the VBS and SOAP experiments during the winter and summer seasons to demonstrate the impact of the mutual use of the 1.5-D VBS scheme and the chosen S/IVOCs parametrizations on the average seasonal absolute impacts. Regarding the winter seasons, Fig. 7a shows that it is mainly manifested by an increase in the average seasonal impacts of emissions from other stationary combustion in the areas with the most significant $PM_{2.5}$ pollution mentioned above, ranging between 1–12 $\mu g\,m^{-3}$ in the Po Valley and mostly between 1–4 $\mu g\,m^{-3}$ in the rest of these areas. Also, this figure reveals that road transport is the only one of the other sectors whose emissions increase the average seasonal impact on $PM_{2.5}$ concentration in the VBS experiment by at least 0.5 $\mu g\,m^{-3}$ in some larger areas. These areas include mainly the Po Valley, where the increase reaches up to 4 $\mu g\,m^{-3}$, as well as parts of southern and western Germany, parts of central Hungary, and parts of southern and central Poland. At the same time, it can be seen that the differences between the average seasonal impacts for the remaining sectors are either small (up to 0.5 $\mu g\,m^{-3}$ in absolute value) or negligible.

Regarding the summer seasons, Fig. 7b indicates that such mutual usage of the 1.5-D VBS scheme and the chosen S/IVOCs parametrizations is mainly associated with an increase in the average seasonal absolute impacts of emissions from road transport in the range of 0.1–2.25 $\mu g\,m^{-3}$ over the entire domain, while the increases exceeding 0.75 $\mu g\,m^{-3}$ occur in southern Poland, roughly in the southern half of Germany, in the north Switzerland and the Po Valley. In addition, it reveals that the average seasonal absolute impacts increase by at least 0.25 $\mu g\,m^{-3}$ only for emissions from other stationary combustion, specifically in the central areas of the Po Valley, where they reach up to 0.75 $\mu g\,m^{-3}$. Finally, it can also be seen that the differences for emissions from the remaining sectors are either smaller than 0.25 $\mu g\,m^{-3}$, especially for those from other stationary combustion, solvents, shipping, and waste, or negligible.

The spatial distributions of the average seasonal absolute contributions of emissions from individual categories (all GNFR sectors, biogenic emissions, initial and boundary conditions) to $PM_{2.5}$ concentration during the winter and summer seasons in the PSAT experiment are illustrated in Fig. 8, while the corresponding spatial distributions of their average seasonal relative contributions are depicted in Fig. S6. During the winter seasons (Figs. 8a and S6a), emissions from other stationary combustion, boundary conditions, road transport, agriculture–livestock, industrial sources, and agriculture–other produce the highest domain-wide absolute seasonal contributions to $PM_{2.5}$ concentration, reaching values of 3.2, 2.1, 1.4, 0.9, 0.6, and 0.5 $\mu g\,m^{-3}$, respectively. The average seasonal contributions of emissions from boundary conditions range between 2–3 $\mu g\,m^{-3}$ in the lower-lying areas of the domain, representing 7.5–30 % of the average seasonal concentration of $PM_{2.5}$. At the same time, these contributions range between 0.5–2 $\mu g\,m^{-3}$ in the higher-lying areas of the domain, representing 25–50 % of the

average seasonal concentration of $PM_{2.5}$. Comparison of the above-mentioned averages for other stationary combustion, road transport, and industrial sources with their corresponding domain-wide averages of the average seasonal impacts in the SOAP experiment, indicating their similarity for other stationary combustion and equality for road transport and industrial sources, is consistent with the striking similarity between the distributions of the average seasonal absolute contributions (Fig. 8a) and the distributions of the average seasonal absolute impacts in the SOAP experiment (Fig. 6a) for these sectors. The same compari-

son for agriculture–livestock and agriculture–livestock, indicating notable differences in their averages, reflects the difference in their corresponding distributions in the PSAT and SOAP experiments, as described in more detail below.

    During the summer seasons (Figs. 8b and S6b), emissions from biogenic sources, road transport, industrial sources, boundary conditions, and other stationary combustion produce the highest domain-wide absolute seasonal contributions to $PM_{2.5}$ concentration, reaching values of 0.57, 0.31, 0.28, 0.27, and 0.25 $\mu g\,m^{-3}$, respectively. Except for the northern, marine, and

highest parts of the domain, the average seasonal contributions of biogenic emissions lie most often between 0.5–1.5 $\mu g\,m^{-3}$, with the highest values being reached in the northwestern region of the Balkan Peninsula. These contributions represent 10–55 % of the seasonal concentration of $PM_{2.5}$. The average seasonal contributions of emissions from boundary conditions reach 0.05–1 $\mu g\,m^{-3}$, with a certain gradient in the northwest direction. Thus, these contributions make up 2.5–30 % of the seasonal concentration of $PM_{2.5}$, with the highest values reached in the Alpine regions.

To quantify the mentioned similarities/differences between the SOAP and PSAT experiments more closely, we plotted the distributions of the difference between the average seasonal impacts in the SOAP experiment and the average seasonal contributions in the PSAT experiment for the individual sectors during the winter and summer seasons in Fig. S7. During the winter seasons (Fig. S7a), the investigated differences are the most pronounced for agriculture–livestock, especially in the lower-lying areas of the domain, where they range between 1.5–4.5 $\mu g\,m^{-3}$. Agriculture–other is the only remaining sector for which these

differences exceed 1 $\mu g\,m^{-3}$, at least on parts of the domain. In the case of other stationary combustion and road transport, they are either negative or positive, depending on the location. The differences for solvents are positive and usually reach up to 0.5 $\mu g\,m^{-3}$, but locally up to 1 $\mu g\,m^{-3}$. For the remaining sectors, the differences are either negligible or slightly negative. During the summer seasons (Fig. S7b), these differences are more pronounced for shipping, road transport, and agriculture–livestock, reaching up to 0.5, 0.75, and 1.25 $\mu g\,m^{-3}$, respectively, especially in the above-mentioned locations, in which the

average seasonal impacts in the SOAP experiment exceed 0.5 $\mu g\,m^{-3}$. For power plants, industrial sources, other stationary combustion, off-road, and agriculture–other, these differences are usually small, the most common to 0.1–0.2 $\mu g\,m^{-3}$. For the remaining sectors (fugitives, solvents, aviation, and waste), they are negligible. Moreover, when comparing the distributions of the differences between the average seasonal impacts and contributions during the winter and summer seasons for $PM_{2.5}$ (Fig. S7) with their counterparts constructed for secondary aerosol (SA; Fig. S8), it is evident that all the above-described

differences for $PM_{2.5}$ are almost exclusively the result of the sum of the contributions formed by the analogous differences for the individual SA components, e.i., for $PNH_4$, $PNO_3$, $PSO_4$, and SOA. For all the sectors, the differences between these distributions for $PM_{2.5}$ and those for SA do not exceed 0.05 $\mu g\,m^{-3}$ in absolute value, with a few exceptions (not shown). In other words, this means that the impacts and contributions are the same for the primary non-reactive components, as expected.

### 3.3.2 Secondary aerosol species

This subsection first deals with the average seasonal contributions of emissions to the individual SA components and then their comparison with their corresponding average seasonal emission impacts. We choose this reverse order here to show, in addition to the seasonal contributions themselves, which of the analyzed emission categories emit the precursor(s) of the given secondary aerosol components, which is directly visible from the seasonal contributions since the PSAT tool is constructed in such a way that each secondary aerosol species is linked only to its direct primary precursor(s), i.e., $PNH_4$ is linked only to $NH_3$, $PNO_3$ to $NO_x$, $PSO_4$ to $SO_2$, and SOA to VOCs and IVOCs (Koo et al., 2009; Burr and Zhang, 2011a). At the same time, because the average seasonal impacts on all the inorganic secondary components in the SOAP experiment are almost identical to their counterparts in the VBS experiment (not shown), only those from the SOAP experiment are presented below.

Figure 9 shows that ammonia emissions from agriculture–livestock and agriculture–other contribute the most to the average seasonal concentration of $PNH_4$ in both seasons. During the winter seasons, the average seasonal absolute contributions of emissions from agriculture–livestock in the Po Valley reach up to $3\ \mu g\,m^{-3}$, while in the rest of the domain up to $0.75–1.25$ $\mu g\,m^{-3}$. The average seasonal absolute contributions of emissions from agriculture–other usually reach $0.5–1\ \mu g\,m^{-3}$. During the summer seasons, the average seasonal absolute contributions of emissions from agriculture–livestock most often range between $0.05–0.7\ \mu g\,m^{-3}$, with values exceeding $0.3\ \mu g\,m^{-3}$ in southern Germany, in the Po Valley, and especially in the northwestern part of Germany. The average seasonal absolute contributions of emissions from agriculture–other reach values between $0.05–0.2\ \mu g\,m^{-3}$ roughly in the northern half of the domain. The average seasonal absolute contributions from the other sectors emitting ammonia are usually smaller, especially for industrial sources, other stationary combustion, fugitives, road transport, and waste in winter seasons, or negligible.

As for $PNO_3$, Fig. 10 indicates that during both seasons, $NO_x$ emissions from boundary conditions contribute the most to its average seasonal concentration over the entire domain, except for the areas in the Po Valley (in the summer seasons also excluding the area of southern Germany). Their average seasonal absolute contributions during the winter seasons reach in the lower-lying areas of the domain $2–3\ \mu g\,m^{-3}$, while in the higher-lying areas, they range between $0.5–2\ \mu g\,m^{-3}$. During the summer seasons, these contributions mostly range between $0.05–1\ \mu g\,m^{-3}$, with values exceeding $0.4\ \mu g\,m^{-3}$ mainly in the northwestern half of Germany. When comparing these results with their counterparts for $PM_{2.5}$, which we mentioned above, it is evident that those specific contributions to $PM_{2.5}$ are formed almost exclusively by $PNO_3$ during both seasons. Further, $NO_x$ emissions from road transport, the second largest contributor to the average seasonal $PNO_3$ concentration over most of the domain in both seasons, are its largest contributor in the central area of the Po Valley during both seasons and in southern Germany during the summer seasons. While their average seasonal contributions range between $3–4\ \mu g\,m^{-3}$ and $0.4–0.8\ \mu g\,m^{-3}$ in the central area of the Po Valley during the winter and summer seasons, respectively, they reach up to $0.6$ $\mu g\,m^{-3}$ in the area of southern Germany during the summer seasons. Other stationary combustion is the last sector whose $NO_x$ emissions contribute to the average seasonal $PNO_3$ concentration during the winter seasons of more than $1.5\ \mu g\,m^{-3}$, namely in the central area of the Po Valley. At the same time, shipping is the last sector whose $NO_x$ emissions contribute to the average seasonal $PNO_3$ concentration during the summer seasons of more than $0.2\ \mu g\,m^{-3}$, namely in northwestern Germany.

The remaining sectors emitting $NO_x$, i.e., power plants, industrial sources, off-road, waste, and agriculture–other, as well as other stationary combustion and shipping in cases different from those previously mentioned, contribute to the seasonal $PNO_3$ concentration less or negligible.

Figure 11a reveals that $SO_2$ emissions from other stationary combustion usually contribute the most to the average seasonal concentration of $PSO_4$ in the winter seasons, especially in the eastern half of the domain, where their average seasonal absolute contributions reach 0.4–1.5 $\mu g\,m^{-3}$. Industrial sources, power plants, and shipping are the remaining sectors whose $SO_2$ emissions in selected domain locations contribute to the average seasonal $PSO_4$ concentration in the winter seasons between 0.1–0.2 $\mu g\,m^{-3}$. At the same time, as can be seen in Fig. 11b, these are the only three sectors whose $SO_2$ emissions in the selected locations of the domain contribute to the average seasonal concentration of $PSO_4$ up to 0.3–0.6 $\mu g\,m^{-3}$ in the summer seasons. In the case of industrial sources and power plants, these locations are mainly in Poland and Germany. Concerning shipping, they are in the Gulf of Venice and Genoa, Italy.

Regarding SOA, Fig. 12a shows that VOCs and IVOCs emissions from other stationary combustion contribute on average the most to the average seasonal concentration of SOA in the winter seasons, with their average seasonal absolute contributions reaching up to 0.4 $\mu g\,m^{-3}$ in the southeastern quarter of the domain and up to 0.8 $\mu g\,m^{-3}$ in the Po Valley. It is also seen that the average seasonal absolute contributions to SOA concentration from the remaining contributing categories, i.e., from solvents, road transport, agriculture–other, biogenic emissions, and boundary conditions, reach up to 0.1–0.2 $\mu g\,m^{-3}$ or are negligible. Further, Fig. 12b reveals that biogenic VOC emissions contribute the most to the average seasonal SOA concentration in the summer seasons. Their average seasonal absolute contributions range between 0.2–1.75 $\mu g\,m^{-3}$, with the highest values reached in the northwestern region of the Balkan Peninsula. Again, when comparing these results with their counterparts for $PM_{2.5}$ mentioned above, it is apparent that those specific contributions to $PM_{2.5}$ during the summer seasons are formed almost exclusively by SOA. Finally, it is also seen that the average seasonal absolute contributions to SOA concentration from the remaining contributing categories, i.e., from other stationary combustion, solvents, road transport, off-road, waste, agriculture– livestock, agriculture–other, and boundary conditions, either reach up to 0.1–0.4 $\mu g\,m^{-3}$ or are negligible.

In order to compare the given average seasonal absolute contributions to the individual secondary aerosol components (Figs. 9–12) with the corresponding average seasonal absolute impacts of emissions on them in the SOAP experiment, we depict these impacts on $PNH_4$, $PNO_3$, $PSO_4$, and SOA during both seasons in Figs. 13–16. Overall, their mutual comparisons indicate that: (1) for sectors that directly emit the precursor(s) of the given secondary component, the distributions of the average seasonal absolute contributions and impacts differ more or less from case to case; (2) the average seasonal absolute impacts of emissions on the given secondary component acquire non-zero and in some cases relatively high or even the highest values even for sectors that do not directly emit its precursor(s) but do emit other precursors that can influence its concentration through the so-called indirect effects, which we deal with in more detail in the discussion. To be precise here, these effects also apply in case (1) if the respective sectors also emit other precursors that can affect the concentration of the respective secondary component.

More specifically, in the case of this comparison for $PNH_4$ (Fig. 9 against Fig. 13), it can be seen that for agriculture– livestock and agriculture–other, the average seasonal absolute impacts over the entire domain are always smaller than the average seasonal absolute contributions in both seasons. Concretely, in the lower areas of the domain, they are smaller up to

0.5–1.5 µg m$^{-3}$ and 0.1–0.5 µg m$^{-3}$ for agriculture–livestock in the winter and summer seasons, respectively. At the same time, they are usually smaller up to 0.5 and 0.1 µg m$^{-3}$) for agriculture–other in the winter and summer seasons, respectively.

On the other hand, mainly for road transport and other stationary combustion, the average seasonal absolute impacts are more or less higher than the average seasonal absolute contributions during the winter seasons, with the highest differences occurring in the central area of the Po Valley, where they reach up to 1 and 0.7 µg m$^{-3}$, respectively. The same is true mainly for power plants, industrial sources, road transport, and shipping during the summer seasons when these differences reach up to 0.2 µg m$^{-3}$ for the first two sectors and up to 0.3 µg m$^{-3}$ for the second two sectors.

The analogous comparison for PNO$_3$ (Fig. 10 against Fig. 14) reveals that the overall highest differences between the average seasonal absolute impacts and contributions during both seasons are associated with emissions from agriculture–livestock, whose average seasonal absolute contributions to PNO$_3$ are 0 µg m$^{-3}$ as agriculture–livestock does not emit NO$_x$. During the winter seasons, the range of these differences is from 0.5 to 6 µg m$^{-3}$, with higher values (above 3 µg m$^{-3}$) observed in the Po Valley and lower values (up to 2 µg m$^{-3}$) observed in higher-lying locations. In the summer seasons, the differences are less

pronounced, ranging from 0.1 to 1.25 µg m$^{-3}$. The highest values are observed in the Po Valley and the northwestern region of Germany. Furthermore, there are also more pronounced differences between the average seasonal absolute for agriculture– other, reaching up to 1–1.5 µg m$^{-3}$ in the winter seasons and up to 0.1–0.25 µg m$^{-3}$ in the summer seasons. At the same time, these differences for power plants, industrial sources, other stationary combustion, road transport, shipping, and off-road are usually small and mostly negative in the winter seasons, whereas they are mostly positive in the summer seasons.

Further, the same type of comparison for PSO$_4$ (Fig. 11 against Fig. 15) shows that during both seasons, the highest differences between the average seasonal absolute impacts and contributions are again related to emissions from agriculture– livestock and agriculture–other, whose average seasonal absolute contributions to PSO$_4$ are 0 µg m$^{-3}$ in both cases since none of them emits SO$_2$. These differences are most pronounced in the eastern half of the domain (in the case of agriculture–livestock also in some areas of Germany), where they locally reach up to 0.3–0.8 µg m$^{-3}$ in the winter seasons and up to 0.2–0.5 µg m$^{-3}$

in the summer seasons. In addition, it can be seen that for power plants, industrial sources, and other stationary combustion (i.e., for sectors that directly emit SO$_2$), the average seasonal absolute impacts are smaller than the average seasonal absolute contributions. This is especially noticeable in the eastern regions (up to 0.25–0.5 µg m$^{-3}$) during the winter seasons. Concerning the average seasonal absolute emission impacts on PSO$_4$ concentration themselves, it is worth mentioning an interesting case in which the reduction of emissions from road transport during the winter seasons causes an increase in the average seasonal

PSO$_4$ concentration, especially over the territory of Poland, by values that exceed its concentration in the base simulation by up to 0.5 µg m$^{-3}$ (Fig. 15a).

    Next, the analogous comparison for SOA (Fig. 12 against Fig. 16) demonstrates that the differences between the average seasonal absolute impacts and contributions during both seasons are usually small (maximally up to $\pm\,0.1$ µg m$^{-3}$), except for those produced by VOCs and IVOCs emissions from road transport. For them, these differences reach up to $\pm\,0.5$ µg m$^{-3}$, with

635 negative values in the areas of the Po Valley during the winter seasons and positive values in scattered areas around the Alps during the summer seasons. Similarly, it is worth mentioning here another interesting case in which the reduction of emissions

from road transport during the winter seasons causes an increase in the average seasonal SOA concentration in the Po Valley by values that exceed its concentration in the base simulation by up to $0.5\ \mu g\,m^{-3}$ (Fig. 16a).

Finally, the comparison of the average seasonal absolute impacts of emissions on SOA in the VBS and SOAP experiments (Fig. 17 against Fig. 16) points that the most substantial differences between them are induced by emissions from other stationary combustion and road transport in the winter seasons, while in the summer seasons, they are caused mainly by emissions from road transport. Specifically, for other stationary combustion in the winter seasons, these differences are particularly pronounced in most of the territory of Czechia, in the Pannonian Basin and its surroundings, where they reach up to 0.8–1.5 $\mu g\,m^{-3}$; however, the highest values, up to $3.5\ \mu g\,m^{-3}$, they reach in the Po Valley. For road transport during the winter seasons, these differences are most pronounced in the Po Valley, where the negative impact on SOA (described above) deepens to values up to $-2\ \mu g\,m^{-3}$. On the other hand, during the summer seasons, these differences for emissions from road transport reach values up to $1.25\ \mu g\,m^{-3}$ in the Po Valley, while in the rest of the domain, they reach values mostly up to 0.5–0.75 $\mu g\,m^{-3}$.

## 3.4 Impacts and contributions in the selected cities

Finally, in this subsection, we present the results connected with assessing the average daily emission contributions to $PM_{2.5}$ concentration in the studied cities and those associated with evaluating the average daily emission impacts on $PM_{2.5}$ concentration in the cities within both sensitivity experiments. Specifically, we focus on describing (1) the sectors whose emissions cause the highest average daily contributions/impacts, which can be seen from Fig. 18–20, and (2) the highest averages of these contributions/impacts in the winter and summer seasons, which are provided in Tables S5–S10.

Figure 18 captures the temporal evolution of the average daily absolute contributions of emissions from all the investigated categories to the concentration of $PM_{2.5}$ in the studied cities within the PSAT experiment. It can be seen that the sums of the average daily absolute contributions from all the categories, representing average daily $PM_{2.5}$ concentrations, are on average higher or even the highest in the late autumn, winter, and early spring months and, conversely, the lowest in the summer months, which is consistent with the annual cycles of average monthly $PM_{2.5}$ concentrations in the cities described during the validation. The highest average daily $PM_{2.5}$ concentrations were reached in Munich ($36.4\ \mu g\,m^{-3}$), Berlin ($41.9\ \mu g\,m^{-3}$), Vienna ($42.2\ \mu g\,m^{-3}$), and Prague ($59.1\ \mu g\,m^{-3}$) during episodes of elevated $PM_{2.5}$ levels in February 2018, while in Budapest ($55.5\ \mu g\,m^{-3}$) and Warsaw ($59.7\ \mu g\,m^{-3}$) during such episodes in December 2018. In contrast, the average daily $PM_{2.5}$ concentrations during the summer months rarely exceed $5\ \mu g\,m^{-3}$ in Berlin, Budapest, and Vienna and $7.5\ \mu g\,m^{-3}$ in Munich, Warsaw, and Prague. As for the highest contributions to the average daily $PM_{2.5}$ concentration, it is seen that they generally occur during the episodes of elevated $PM_{2.5}$ levels in all the studied cities, especially in the winter months. Moreover, except for Munich, the highest average daily contributions are caused by emissions from other stationary combustion. These contributions reach up to 11, 15, 26, 27.5, and $30\ \mu g\,m^{-3}$ in Berlin, Vienna, Warsaw, Prague, and Budapest, respectively. In Munich, the highest average daily contributions, which reach up to $8.9\ \mu g\,m^{-3}$, are caused by emissions from road transport, while emissions from other stationary combustion can produce the second highest contributions there. These contributions reach up to $8.6\ \mu g\,m^{-3}$. Emissions from road transport are the second largest contributor in all the other cities studied except Munich. Their

contributions reach up to 16 $\mu g\,m^{-3}$ in Prague and up to 8–10.5 $\mu g\,m^{-3}$ in the other cities. The third highest contributions, which exceed 5 $\mu g\,m^{-3}$, are produced by emissions from agriculture–other in Berlin, Munich, Vienna, and Prague, while they are caused by emissions from agriculture–livestock in Budapest and by emissions from industrial sources in Warsaw. Regarding the seasonal averages of the average daily absolute/relative contributions to $PM_{2.5}$ concentration for the winter seasons, Tables S5–S10 show that: (1) in all the cities, the three highest ones are caused by emissions from other stationary combustion, road transport, and boundary conditions; (2) the highest ones are caused by emissions from boundary conditions in Berlin and Munich, while in other cities, they are generated by emissions from other stationary combustion. Concerning the similar seasonal averages for the summer seasons, the mentioned tables show that among the three highest are those caused by emissions from industrial sources, road transport, other stationary combustion, or biogenic emissions, depending on the specific city. At the same time, it can be seen that, except for the seasonal average caused by emissions from road transport in Prague, they do not exceed 1 $\mu g\,m^{-3}$ and 30 %, respectively.

The temporal evolution of the average daily absolute impacts of emissions from individual GNFR sectors on the concentration of $PM_{2.5}$ in the studied cities within the SOAP experiment is shown in Fig. 19. When comparing it with Fig. 18, it can be seen that the sums of the average daily impacts in each of the cities almost copy the temporal evolution of the sums of the average daily contributions. The Pearson correlation coefficient between them reaches a minimum value of 0.97 in all the cities. The total differences between the average daily impacts from the SOAP experiment and the average daily contributions from the PSAT experiment caused by emissions from all the anthropogenic sources (i.e., in the sense of the sum of these differences from all the anthropogenic sources) are almost always positive throughout both years in all the studied cities (Fig. S9). Moreover, these differences acquire the highest values during the autumn and winter months when they reach 11, 11.4, 12.9, 13.4, 16.3, and 19.5 $\mu g\,m^{-3}$ in Vienna, Berlin, Budapest, Warsaw, Prague, and Munich, respectively. At the same time, Fig. S9 demonstrates that these differences are mainly caused by emissions from agriculture–livestock. Figure 19 further reveals that agriculture–livestock, other stationary combustion, and road transport are the three sectors whose emissions cause the highest daily impacts in Berlin, Munich, and Prague, while other stationary combustion, agriculture–livestock and agriculture–other are such sectors in Budapest, Vienna, and Warsaw. At the same time, the highest average daily impacts are caused by emissions from agriculture–livestock in Berlin and Munich, in which they reach up to 17.3 and 19.7 $\mu g\,m^{-3}$, respectively. On the other hand, the highest average daily impacts are produced by emissions from other stationary combustion in Vienna, Warsaw, Prague, and Budapest, in which they reach up to 17.2, 23, 29.7, and 30.4 $\mu g\,m^{-3}$, respectively. In connection with the seasonal averages of the average daily absolute/relative impacts on $PM_{2.5}$ concentration for the winter seasons, Tables S5–S10 reveal that: (1) in all the cities, the three highest ones are caused by emissions from other stationary combustion, agriculture–livestock and road transport; (2) the highest ones are caused by emissions from agriculture–livestock in Berlin and Munich, while in the other cities, they are produced by emissions from other stationary combustion. As regards the seasonal averages of the average daily absolute/relative impacts for the summer seasons, Tables S5–S10 show that among the three highest are those caused by emissions from sectors industrial sources, road transport, agriculture–livestock, and other stationary combustion, depending on the specific city. At the same time, it can be seen that, except for the seasonal average caused by emissions from road transport in Prague, they do not exceed 1.1 $\mu g\,m^{-3}$ and 33 %, respectively.

Figure 20 depict the temporal evolution of the average daily absolute impacts of emissions from individual GNFR sectors on the concentration of $PM_{2.5}$ in the studied cities within the VBS experiment. When comparing it with Fig. 19, it can be seen that the sums of the average daily impacts from both sensitivity experiments follow nearly the same temporal pattern in each of the cities. The Pearson correlation coefficient between them exceeds a value of 0.99 in all the cities. The total differences between the average daily impacts from the VBS and SOAP experiments produced by emissions from all the anthropogenic sources (again, in the sense of the sum of these differences from all the anthropogenic sources) are positive throughout both years in all the cities (not shown). Moreover, these differences achieve the highest values in the winter months, during which they reach up to 5.2, 6.2, 7.1, 11.5, 15.8, and 17.7 $\mu g\,m^{-3}$ in Munich, Berlin, Vienna, Warsaw, Prague, and Budapest, respectively. At the same time, Fig. S10 shows that emissions from other stationary combustion predominantly produce these differences; however, emissions from road transport also strongly influence them in Berlin and Munich. Figure 20 further reveals that the three sectors whose emissions cause the highest daily impacts in the individual cities are the same as those in the abovementioned SOAP experiment. Furthermore, it shows that emissions from agriculture–livestock produce the highest average daily impacts in Berlin and Munich, in which they reach up to 17.4 and 19.8 $\mu g\,m^{-3}$, respectively. Also, it can be seen that emissions from other stationary combustion caused the highest average daily impacts in Vienna, Warsaw, Prague, and Budapest, in which they reach up to 22.4, 29.4, 41.5, and 45.1 $\mu g\,m^{-3}$, respectively. Regarding the seasonal averages of the average daily absolute/relative impacts on $PM_{2.5}$ concentration for the winter seasons, Tables S5–S10 reveal that: (1) in all the studied cities, the three highest ones are caused by emissions from other stationary combustion, agriculture–livestock, and road transport; (2) the highest ones are produced by emissions from agriculture–livestock in Berlin and Munich, while in the other cities, they are caused by emissions from other stationary combustion. As regards the seasonal averages of the average daily absolute/relative impacts for the summer seasons, Tables S5–S10 show that among the three highest are, depending on the specific city, those caused by emissions from industrial sources, road transport, agriculture–livestock, and other stationary combustion, while the highest ones are produced in all the cities by emissions from road transport.

In order to provide a complete picture of which $PM_{2.5}$ components are responsible for the differences between the average seasonal impacts in the SOAP experiment and the average seasonal contributions in the PSAT experiment, as well as between the average seasonal impacts in the VBS and SOAP experiments, at the level of the studied cities presented in Tables S5–S10, we show the corresponding average seasonal impacts and contributions for individual modeled $PM_{2.5}$ components in Tables S11–S17. Specifically, Tables S11–S13 show them for three primary components, i.e., primary elemental carbon (PEC), fine primary another inorganic aerosol (FPRM), and POA, respectively. Tables S14–S17 show them for the secondary components, i.e., for $PNH_4$, $PNO_3$, $PSO_4$, and SOA, respectively. Overall, the results of this extended analysis are in complete agreement with those arising from the spatial distributions over the areas of the individual cities. Tables S11–S13 confirm that the impacts are equal to the contributions for primary chemically non-reactive components. The differences between the average seasonal impacts in the VBS and SOAP experiments are mainly attributed to POA during the winter seasons (Table S13). At the same time, they are attributed to both SOA and POA during the summer seasons (Table S13 and S17). Finally, Table S15 shows that even in all the cities studied, the most prominent difference between the average seasonal contributions and impacts is associated with the indirect effect of emissions from agriculture–livestock on $PNO_3$ during the winter seasons.

## 4 Discussion and conclusions

In this work, we focused on analyzing activity sources of fine PM and its secondary components (with an emphasis on sources from anthropogenic activity) in the region of Central Europe using two different approaches applied within the framework of chemical transport modeling. In the first case, we used an extreme case of the brute-force method, the so-called zero-out method, to determine the impacts of a complete reduction of emissions from individual anthropogenic activities on fine PM and its secondary components. In addition, we tested the impact of the implementation of the organic aerosol chemistry/partitioning, together with the inclusion of I/SVOCs emissions estimates, on the changes in the mentioned impacts. In the second case, we used the PSAT tool to determine the contributions of emissions from individual anthropogenic activities to fine PM and its secondary components. At the same time, we compared the outcomes, i.e., the impacts and contributions, resulting from both of these approaches.

Before discussing the chemical part of the validation, we consider it appropriate to briefly discuss the part devoted to the meteorological elements. The comparison of the average modeled annual and diurnal air temperature cycles with those measured over Prague showed that the WRF model can capture them quite accurately. Moreover, the biases between these diurnal cycles in both seasons are very similar to those determined by Liaskoni et al. (2023) when comparing the simulation performed by the WRF model on a similar domain with the same horizontal resolution, albeit with the different settings of parametrizations settings, at 10 Czech stations for the period 2007–2016. As for the wind speed, we showed that WRF in our setting overestimates the average annual cycle over Prague except for the summer months throughout the year, with the most substantial overestimation occurring during the winter months. This result is consistent with the results of the validation performed by Karlický et al. (2020), who showed a positive bias of the modeled average seasonal wind speeds predicted by the WRF model on the Central European domain with a similar horizontal resolution (10 km) in its multiple different settings both during winter and summer, with more pronounced modeled overestimation during winter. The overestimation of wind speed by the WRF model was shown or mentioned in several other studies (e.g., Terrenoire et al., 2015; Huszar et al., 2020a; Liaskoni et al., 2023). Such an overestimation can represent a potential source of the underestimation of $PM_{2.5}$ concentrations in our simulations. For example, Aksoyoglu et al. (2011) achieved an increase in PM concentrations by a factor of 2–3 when they reduced modeled wind speeds during observed periods of low wind.

Regarding the chemical part of the validation, we first presented the comparison between the modeled and measured $PM_{2.5}$ concentrations in the selected cities of the studied region (Berlin, Munich, Vienna, Budapest, Warsaw, and Prague), which, among other things, confirmed the high consistency of the CAMx model in predicting $PM_{2.5}$ concentrations with and without using the PSAT tool. At this point, it is worth noting that the subtle difference between the base simulation of the SOAP experiment and the simulation of the PSAT experiment stems from the different precision of emission fluxes in FUME and CAMx that is next transferred as a result of numerical rounding to the subtle differences in the total emissions used in the SOAP and PSAT experiments; however, these differences are small or negligible, and have no substantial effect on the results related to emission contributions/impacts on $PM_{2.5}$ and its components. In addition to this consistency, the comparison showed that the use of the 1.5-D VBS scheme together with the estimates of I/SVOCs emissions leads to a slight improvement of the

775 overall model prediction of $PM_{2.5}$ in the studied cities, i.e., when taking into account all seasons/months of the year, even if they can slightly deteriorate it in some cases. This improvement results from the fact that when using the 1.5-D VBS scheme together with I/SVOCs emissions, there is an increase in average $PM_{2.5}$ concentrations compared to those modeled by the SOAP scheme (Figs. 2 and 5), which in both cases are mostly underestimated compared to the measurements. The increase in average $PM_{2.5}$ concentrations is almost exclusively due to the rise in POA and SOA concentrations (Fig. S11 and Tables

S11–S17). Such an improvement in the model prediction of $PM_{2.5}$ when using the 1.5-D VBS scheme or its modifications together with additional I/SVOCs emissions is expected since their implementation typically leads to an improvement in the prediction of organic aerosol (Ciarelli et al., 2017; Giani et al., 2019; Jiang et al., 2019b, 2021). At the same time, however, it is necessary to add that the current implementation of this concept is burdened by several uncertainties and therefore requires additional revisions that can further improve the model prediction of organic aerosol and thus the total fine PM. We refer to the

articles above for a more detailed description of some of the uncertainties mentioned.

In connection with our validation of $PM_{2.5}$ concentrations in the selected cities, Liaskoni et al. (2023) performed a similar comparison in the same cities but for the period 2007–2016. To model $PM_{2.5}$, they used the same version of the CAMx model on a similar domain with the same horizontal resolution but with slightly different settings, driving meteorological fields obtained by the WRF model, and older emission inputs. We note that their settings in the simulation without wind-blown dust

emissions and realized using the ISORROPIA module mainly correspond to those we used in the base simulation of the SOAP experiment. In general, the seasonal correlations and NMBs determined by us are in reasonable qualitative agreement with those presented by them: (1) the seasonal correlations are mostly the highest during winter and the lowest during summer; (2) the modeled concentrations are on average underestimated the most during summer, while the greatest match between the modeled and measured concentrations occurs in the cold half-year (October–March). Further, Huszar et al. (2021) also

compared modeled and measured average monthly concentrations of $PM_{2.5}$ in these cities, however, for an earlier period (2015–2016). To model $PM_{2.5}$, they also used the CAMx model (albeit in an older version) on a similar domain in the same horizontal resolution but with a slightly different setting (corresponding again mainly to those we used in the base simulation of the SOAP experiment), different meteorological fields obtained by the WRF model, and older emissions. Despite this, the mutual relations of the average annual cycles of the monthly $PM_{2.5}$ concentrations determined by them in most of the studied

cities show qualitatively similar patterns as in our case. The same applies when comparing analogous cycles in these cities, which were reported by Liaskoni et al. (2023).

The comparison of the average annual cycles of monthly $PM_{2.5}$ concentrations over the rural and (sub)urban stations revealed that the CAMx model underestimates both during the year. Qualitatively, the same results were also found by Huszar et al. (2024), who modeled $PM_{2.5}$ for the period 2015–2016 using the CAMx model in a very similar experimental setup to the one

we used in the base simulation of the SOAP experiment, but with the difference that CAMx was driven by the regional climate model RegCM version 4.7 (Giorgi et al., 2012). Further, we found that all the average modeled annual cycles of monthly $NO_2$ concentrations are systematically underestimated. Huszar et al. (2016, 2020a, 2021) found qualitatively the same results as well. Huszar et al. (2020a) suggested underestimation of $NO_2$ emissions or at least a problem with the speciation of $NO_x$ emissions into NO and $NO_2$ as possible causes of these underestimations. Due to the remarkable similarity of experimental

setups and emission preprocessing in their and our experiments, the reasons given are also relevant to the underestimations found in our simulations. Concerning $SO_2$, we found that the model often fails to capture the average annual cycles of its monthly concentrations in the studied cities. The same fact was also pointed out by Huszar et al. (2022), who mentioned deficiencies in the annual profile used to time-disaggregate annual emissions to monthly ones and wrong vertical turbulent mixing as possible reasons for that. Because we used the same methods for time disaggregation and calculating the vertical eddy-diffusion turbulent coefficients, these factors may also play an important role in our simulations.

The comparison of the modeled and measured average annual cycles of $PM_{2.5}$ components showed that the main components responsible for the model underestimation of $PM_{2.5}$ throughout the year are mainly OC, followed by $PSO_4$. Interestingly, the relationships between the average modeled and measured annual cycles of monthly $PNH_4$, $PNO_3$, and $PSO_4$ concentrations that we found are qualitatively the same as those found by Huszar et al. (2024) but differ quantitatively. The quantitative differences might be associated with the different meteorological drivers used (WRF vs. RegCM), while the qualitative similarities might indicate problems with emissions. Based on the mentioned similarity, the great underestimation of $PSO_4$ in our simulations during the cold half-year may be related to the overestimation of $PNO_3$, which consumes available $NH_3$ and suppresses the formation of $PSO_4$, similarly as suggested by Huszar et al. (2024). The underestimation of $PSO_4$ during the warm half-year could be related to the factors affecting the annual cycle of $SO_2$, which we mentioned above. Among the potential sources of uncertainty causing the underestimation of organic aerosol are (1) uncertainties in its emission inventories, which is partially consistent with the missing I/SVOCs emissions discussed above, and (2) estimates of emissions of biogenic volatile organic compounds, especially in the warm half-year, as they can significantly affect SOA concentrations (Jiang et al., 2019a). Other important sources of uncertainty in modeled $PM_{2.5}$ concentrations in some regions of Central Europe could be wind-blown dust emissions, especially in the cold half-year (Liaskoni et al., 2023).

The crucial conclusion of the model evaluation, i.e., more or less significant underestimation of modeled $PM_{2.5}$ concentrations with a few exceptions, must be considered when interpreting all the results concerning the contributions and impacts of emission sources. Specifically, it can be assumed that the average absolute contributions and impacts determined for $PNO_3$ and $PNH_4$ during the winter seasons are on average slightly overestimated, while those for $PSO_4$ and especially those for organic aerosol are on average slightly underestimated both in the winter and summer seasons. Overall, it can be assumed that the average absolute contributions and impacts determined for $PM_{2.5}$ are on average slightly underestimated.

As we already mentioned in the introduction, Pültz et al. (2023) used the LOTOS-EUROS model on a domain with a similar horizontal resolution as we used in our experiments to determine the average annual concentration of $PM_{2.5}$ (10.4 $\mu g\,m^{-3}$) as well as the average annual contributions of emission sectors in Berlin during the period 2016–2018. These contributions for other stationary combustion, power plants and industrial sources, boundary conditions, agriculture–livestock and agriculture–other, road transport, biogenic emissions, and other sectors are 3.2, 2.0, 1.4, 1.3, 1.3, 0.5, and 0.7 $\mu g\,m^{-3}$, respectively. Considering the relatively small time difference between their and PSAT experiments, we can assume a mutual similarity of their results with the counterparts determined from the PSAT experiment. In order to compare them, we determined these counterparts: the average annual concentration of $PM_{2.5}$ is 6.8 $\mu g\,m^{-3}$ and the average annual contributions of the emission sectors are 1.0, 1.1, 1.3, 0.9, 1.1, 0.3, and 1.1 $\mu g\,m^{-3}$, respectively. Contrary to our assumption, the average annual $PM_{2.5}$ concentra-

845 tion in the PSAT experiment is underestimated by a factor of around 1.53. Moreover, it is evident that this underestimation is mainly caused by underestimating the contribution of emissions from other stationary combustion by a factor of 3.2, followed by underestimating the contribution of emissions from power plants and industrial sources by a factor of around 1.82. These observed differences could be partially explained by the use of different emission databases for the territory of Germany in both experiments. While we used emissions from the CAMS database, Pültz et al. (2023) applied gridded emissions obtained

from the GRETA (Gridding Emission Tool for ArcGIS v1.1; Schneider et al., 2016) system with the exception of emissions for residential wood combustion (RWC), which they replaced with a scientific bottom-up inventory accounting for the semivolatile components of these emissions (Denier van der Gon et al., 2015). Thus, they used the RWC emissions increased compared to those officially reported in the GRETA system by a factor of 2–3, which is naturally reflected in the average annual contribution of emissions from other stationary combustion since the RWC emissions contribute to this sector. The fact that the emissions

for RWC reported in the CAMS database also do not consider the presence of semivolatile compounds could partially explain the observed largest underestimation of the annual contribution of emissions from other stationary combustion in the PSAT experiment.

  When comparing the total monthly contributions of emissions to $PM_{2.5}$ in Berlin, Budapest, and Warsaw determined for February and August 2010 by Karamchandani et al. (2017), we found that compared to our determined total seasonal contribu-

860 tions to $PM_{2.5}$ in the winter and summer seasons, they are higher by factors of 1.7–3.0 and 3.0–3.9, respectively. The decrease in the total contributions determined by us could be partly explained by the reduction in anthropogenic emissions over the course of 9 years (Karamchandani et al. (2017) used the TNO-MACC_II emission inventory (Kuenen et al., 2014) for the year 2009). However, differences in other factors, such as the spatial resolution of model experiments, driving meteorological fields, or other emission inputs, should also participate in it. Also, the inconsistency of comparing the total monthly and seasonal

contributions can play a role. A deeper qualitative comparison between the compositions of the contributions from the individual sectors determined by them and us shows the persistent dominance of the contributions from other stationary combustion, followed by the contributions from road transport, public power and industry, and agriculture in all three cities during winter. It is appropriate to mention here that we considered the mutual influence of emissions from power plants and industrial sources as we used different nomenclature of anthropogenic sectors, which made it difficult to distinguish between these two sectors

in our and their work. For the same reason, we considered the mutual influence of emissions from agriculture–livestock and agriculture–other. In contrast to our findings, in their case, emissions from boundary conditions do not appear among the most significant contributions during winter, while this is the case during summer. The observed discrepancy during winter could partly be explained by the fact that they used a model domain extending over Europe. Thus, the contributions of anthropogenic emissions released from European regions outside our domain are included directly in their determined contributions from

individual anthropogenic sectors. The fact that we did not include dust emissions in the PSAT experiment, which Karamchandani et al. (2017), on the other hand, considered in the domain and boundary conditions framework, could somewhat clarify the observed discrepancy between the contributions during summer. Overall, the contributions found by us and them during the summer are less consistent than those during the winter.

Regarding the differences between the contributions and impacts determined for $PM_{2.5}$ during both studied seasons, we have shown that they were generated almost exclusively by secondary aerosol components. This conclusion fully agrees with the results of Koo et al. (2009), who showed excellent agreement between the contributions and impacts determined for primary $PM_{2.5}$. As they argue, this is to be expected because the source–receptor relationships for primary PM are essentially linear and not affected by any indirect effects. The same argumentation can be used in our case as well. Moreover, Koo et al. (2009), as well as Burr and Zhang (2011a, b) who applied the same methods as we did to determine the contributions and impacts of emissions above the eastern United States for January and July 2002, shed light on the general principles (along with specific examples) explaining the essence of the differences between the two approaches. These differences are caused by the acting of (1) oxidation–limiting effects in the perturbed and base simulation of sensitivity experiments as well as in a simulation with the applied PSAT mechanism and/or (2) indirect effects, which are not considered when using the PSAT mechanism, in the perturbed simulation. An indirect effect is generally an effect in which a change in the concentration of a specific secondary aerosol component is conditioned by a modification in the emissions of its indirect gaseous precursor(s). The PSAT mechanism, as we have already mentioned and also shown when evaluating the seasonal contributions of secondary aerosol components, assigns contributions to a specific secondary aerosol component (e.g., $PNH_4$) only to sectors (sources) that emit its direct precursor(s) (i.e., $NH_3$), and thus considers only direct effects. As an example of the indirect effect, we mention a decrease in the concentration of $PNO_3$ caused by a significant reduction in the emissions of $NH_3$ from agriculture–livestock (its dominant source), which limits the production of ammonium nitrate ($NH_4NO_3$), leaving more $HNO_3$ in the gas phase. This decrease in the concentration of $PNO_3$ in the perturbed simulation is naturally reflected in the values of the determined daily/seasonal average emission impacts of emissions from this sector on $PNO_3$, which in turn are mainly responsible for the overall highest differences between the daily/seasonal contributions to $PM_{2.5}$ and their corresponding impacts found among all the anthropogenic sectors just for agriculture–livestock (Figs. S7 and S9, Tables S5–S10 and S15). For a more detailed description of other indirect and oxidation–limiting effects, with the help of which it is possible in principle to clarify other observed differences between the contributions and impacts in our work, we refer to the articles mentioned above.

The main conclusions about the contributions/impacts of emissions to/on the concentrations of fine PM and its secondary components, established in this paper for the region of Central Europe and the selected large cities, can be briefly summarized as follows:

– In general, the average seasonal/daily absolute/relative contributions of emissions to the concentration of $PM_{2.5}$ and its secondary components are strongly spatially and temporally conditioned. The same goes for their corresponding impacts.

– In the winter seasons, the average seasonal absolute contribution from other stationary combustion dominates most of the region's territory except for its western areas, followed by emissions from boundary conditions, road transport, agriculture–livestock, industrial sources, and agriculture–other. Their domain-wide averages are 3.2, 2.1, 1.4, 0.9, 0.6, and 0.5 $\mu g\,m^{-3}$, respectively. In the summer seasons, the average seasonal absolute contribution from biogenic emissions dominates most of the region's territory, followed by emissions from road transport, industrial sources, boundary conditions, and other stationary combustion. Their domain-wide averages of 0.57, 0.31, 0.28, 0.27, and 0.25 $\mu g\,m^{-3}$,

respectively. The highest daily contributions to the average daily $PM_{2.5}$ concentration, occurring during episodes of elevated $PM_{2.5}$ levels in all the cities, especially in the winter months, are predominantly produced by emissions from other stationary combustion, followed by emissions from road transport. The three highest seasonal averages of the average daily absolute contributions to $PM_{2.5}$ concentration during the winter seasons in all the cities are caused by emissions from other stationary combustion, road transport, and boundary conditions, with the order depending on the specific city. During the summer seasons, they are caused by emissions from industrial sources, road transport, other stationary combustion, or biogenic emissions, depending on the specific city. The main contributors to the average seasonal concentration of $PNH_4$ in both seasons are $NH_3$ emissions from agriculture–livestock and agriculture–other. $NO_x$ emissions from boundary conditions and road traffic are the main contributors to the average seasonal concentrations of $PNO_3$ in both seasons. The main contributors to the average seasonal concentration of $PSO_4$ during the winter seasons are $SO_2$ emissions from other stationary combustion, power plants, and industrial sources, while during the summer seasons, they are mainly emissions from power plants, industrial sources, and shipping. Finally, VOC and IVOCs emissions from other stationary combustion are the main contributors to the average seasonal concentration of SOA during winter seasons, while BVOCs emissions are such contributors during the summer seasons.

– In contrast, the most enormous average seasonal absolute impacts on $PM_{2.5}$ concentration caused by anthropogenic emissions in the SOAP experiment during the winter seasons are those from other stationary combustion, agriculture–livestock, road transport, agriculture–other, and industrial sources. Their domain-wide averages are 3.4, 2.9, 1.4, 1.1, and 0.6 $\mu g\,m^{-3}$, respectively. During the summer seasons, among such impacts are those from agriculture–livestock, road transport, industrial sources, other stationary combustion, and shipping. Their domain-wide averages are 0.46, 0.45, 0.34, 0.29, and 0.20 $\mu g\,m^{-3}$, respectively. Further, the sectors whose emissions cause the highest daily impacts on $PM_{2.5}$ concentration in the cities are primarily other stationary combustion and agriculture–livestock, followed by road transport or agriculture–other, with their specific order depending on the specific city. The three highest seasonal averages of the average daily impacts on $PM_{2.5}$ concentration during the winter seasons in the cities are rendered by emissions from other stationary combustion, agriculture–livestock, and road transport, while among the three highest such averages during the summer seasons are those generated by emissions from industrial sources, road transport, agriculture–livestock, and other stationary combustion, depending on the specific city.

– The differences between the contributions of emissions from anthropogenic sectors to $PM_{2.5}$ concentration in the PSAT experiment and the impacts of these emissions on $PM_{2.5}$ concentration in the SOAP experiment are predominantly induced by the acting of oxidation–limiting and/or indirect effects on secondary aerosol components. The most substantial of these differences are associated with emissions from agriculture–livestock, mainly due to the differences in particulate nitrate concentrations. The highest differences in these concentrations reach in terms of daily averages up to around 15 $\mu g\,m^{-3}$ in some of the studied cities during wintertime and in terms of seasonal averages up to 4.5 and 1.25 $\mu g\,m^{-3}$ in the winter and summer seasons, respectively.

– Finally, modeling of gas-aerosol partitioning and chemical aging of organic aerosol using the 1.5-D VBS scheme and including the estimations of I/SVOCs emissions within the VBS experiment, compared to the use of the SOAP scheme, is mainly manifested by an increase in the average seasonal impacts on the concentration of PM$_{2.5}$ caused by emissions from other stationary combustion and road transport during the winter seasons and by emissions from road transport during the summer seasons. These increases reach up to 12 and 4 $\mu g\,m^{-3}$, respectively, during the winter seasons and up to 2.25 $\mu g\,m^{-3}$ during the summer seasons. Qualitatively, the same conclusions also apply to increases in the daily averages in the cities.

The results presented in this paper provide detailed and valuable information about the contributions of emissions from a broad spectrum of anthropogenic activities to the current composition of fine PM in Central Europe and its selected metropolises, as well as about the impacts of potential overall emission reductions within individual activity sectors on its composition. These can be used, at least as framework estimates, in designing appropriate strategies to reduce this kind of air pollution.

The above-discussed possible reasons leading to the shortcomings of the model system used in capturing the concentration of fine PM indicate our future activities to eliminate them potentially, such as the inclusion of dust emissions within the scope of the domain and boundary conditions. In addition, an inherent aspect in the effort to improve the overall quality of model experiments will be a significant increase in their resolution, at least as additional nested domains covering selected areas of interest, e.g., selected urban areas.

## Appendix A:  Definitions of average temporal impacts and contributions

Based on the principle of the zero-out method, we define the average temporal absolute impact of emissions from the sector of anthropogenic activity $x$ on the concentration $c(i)$ of chemical species (or their aggregate) $i$ as:

$$\overline{I_x^{\mathrm{abs}}(c(i))} = \frac{1}{N} \sum_{j=1}^{N} \left( c_j^{\mathrm{BASE}}(i) - c_j^x(i) \right), \tag{A1}$$

where $c_j^{\mathrm{BASE}}(i)$ and $c_j^x(i)$ are the average hourly concentrations of chemical species (or their aggregate) $i$ in the base simulation and the perturbed simulation with zero emissions from sector $x$, respectively, falling within the appropriate time interval, and $N$ is their total number. The average temporal relative impact of emissions from the sector of anthropogenic activity $x$ on the concentration of PM$_{2.5}$ is considered as:

$$\overline{I_x^{\mathrm{rel}}(\mathrm{PM}_{2.5})} = 100 \, \frac{\overline{I_x^{\mathrm{abs}}(\mathrm{PM}_{2.5})}}{\frac{1}{N} \sum_{j=1}^{N} c_j^{\mathrm{BASE}}(\mathrm{PM}_{2.5})}, \tag{A2}$$

where $\overline{I_x^{\mathrm{abs}}(\mathrm{PM}_{2.5})}$ is calculated by Eq. (A1), $c_j^{\mathrm{BASE}}(\mathrm{PM}_{2.5})$ are the average hourly concentrations of PM$_{2.5}$ in the base simulation falling within the appropriate time interval, and $N$ is their total number. Thus, Eq. (A2) shows that $\overline{I_x^{\mathrm{rel}}(\mathrm{PM}_{2.5})}$ defined by us represents the ratio between $\overline{I_x^{\mathrm{abs}}(\mathrm{PM}_{2.5})}$ and the corresponding time-averaged concentration of PM$_{2.5}$ in the base simulation, expressed as a percentage.

In connection with the source apportionment given by the PSAT tool, we define the average temporal absolute contribution of emissions from the given category $x$ to the concentration $c(i)$ of chemical species (or their aggregate) $i$ as:

$$\overline{C_x^{\text{abs}}(c(i))} = \frac{1}{N} \sum_{j=1}^{N} c_j^x(i), \tag{A3}$$

where $c_j^x(i)$ are the average hourly concentrations of chemical species (or their aggregate) $i$ allocated to the given category $x$ by the PSAT tool that fall within the appropriate time interval, and $N$ is their total number. It is worth mentioning here that the allocation in the PSAT experiment is split into 15 categories, represented by individual GNFR sectors A–L, biogenic emissions, initial condition, and boundary conditions. Finally, the average temporal relative contribution of emissions from the given category $x$ to the concentration of $PM_{2.5}$ is considered as:

$$\overline{C_x^{\text{rel}}(PM_{2.5})} = 100 \frac{\overline{C_x^{\text{abs}}(PM_{2.5})}}{\overline{C_{\text{tot}}^{\text{abs}}(c(PM_{2.5}))}}, \tag{A4}$$

where $\overline{C_x^{\text{abs}}(PM_{2.5})}$ is calculated by Eq. (A3), and $\overline{C_{\text{tot}}^{\text{abs}}(c(PM_{2.5}))}$ represents the sum of $\overline{C_x^{\text{abs}}(PM_{2.5})}$ over all 15 above mentioned categories. Thus, Eq. (A4) illustrates that $\overline{C_x^{\text{rel}}(PM_{2.5})}$ defined by us represents the ratio between $\overline{C_x^{\text{abs}}(PM_{2.5})}$ and the corresponding time-averaged concentration of $PM_{2.5}$ in the PSAT experiment, expressed as a percentage.

*Code and data availability.* CAMx version 7.10 is available at http://camx-wp.azurewebsites.net/download/source (Ramboll, 2022). The WRF version 4.2 used in the study is available at https://github.com/wrf-model/WRF/releases (WRF, 2023). The observational data from the AirBase database can be obtained from https://discomap.eea.europa.eu/map/fme/AirQualityExport.htm. (EEA, 2023). The CAMS emission data can be obtained from https://permalink.aeris-data.fr/CAMS-REG-ANT (Kuenen et al., 2021). The Czech REZZO and ATEM emission data can be obtained upon request from their publishers, the Czech Hydrometeorological Institue (https://www.chmi.cz) and the Studio of Ecological Models (https://www.atem.cz). The complete model configuration and all the simulated data (3-dimensional hourly data) used for the analysis are stored at the Dept. of Atmospheric Physics of the Charles University data storage facilities (about 3TB) and are available upon request from the main author.

*Author contributions.* LB, KE, and JK performed the model simulations; LB and PH contributed to the data analysis and writing of the manuscript; OV conceptualized the study and planned the experiments.

*Competing interests.* No competing interests are present.

*Acknowledgements.* This work has been supported by the Czech Technological Agency (TACR) grant No.SS02030031 ARAMIS (Air Quality Research Assessment and Monitoring Integrated System) and Charles University SVV 260709 project.

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

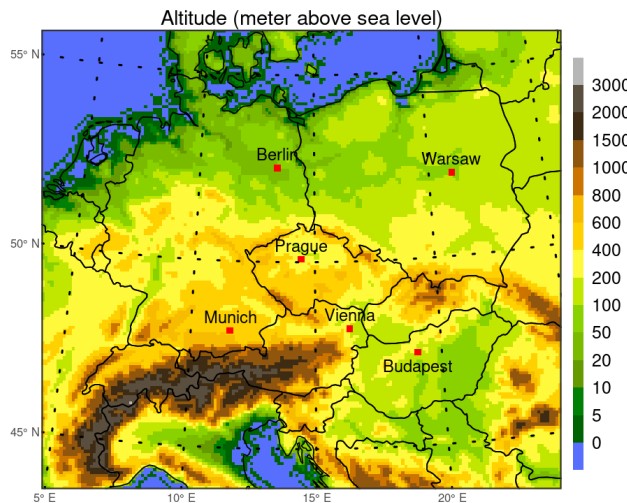

**Figure 1.** The resolved model terrain altitude (in meters above sea level) and the locations of the cities analyzed in the study (Prague, Berlin, Munich, Vienna, Budapest, Warsaw).

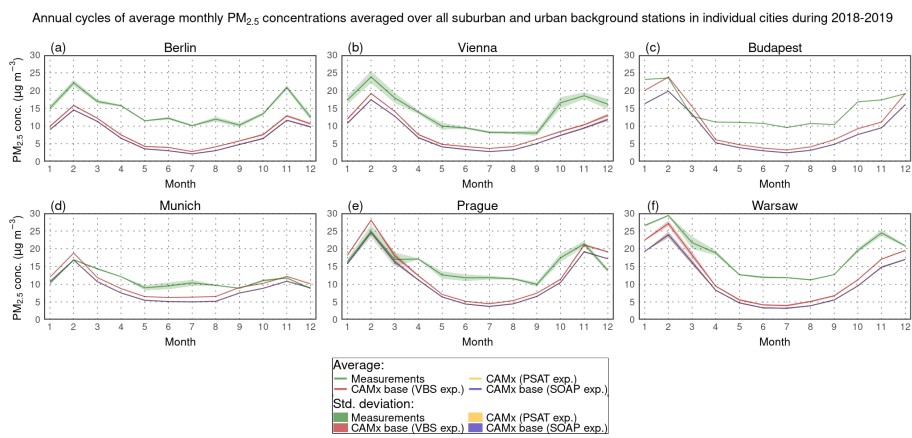

**Figure 2.** Comparison of modeled (the base simulation of the SOAP/VBS experiment – blue/red lines, the simulation of the PSAT experiment – orange lines) and measured (AirBase data – green lines) annual cycles of average monthly PM$_{2.5}$ concentrations (in µg m$^{-3}$) averaged over all suburban and urban background stations in Berlin **(a)**, Vienna **(b)**, Budapest **(c)**, Munich **(d)**, Prague **(e)**, and Warsaw **(f)** during 2018–2019. The colored areas indicate the standard deviations of the averages, calculated using Eq. (S4) provided in the Supplement. Their color scale corresponds to the scale used for the averages.

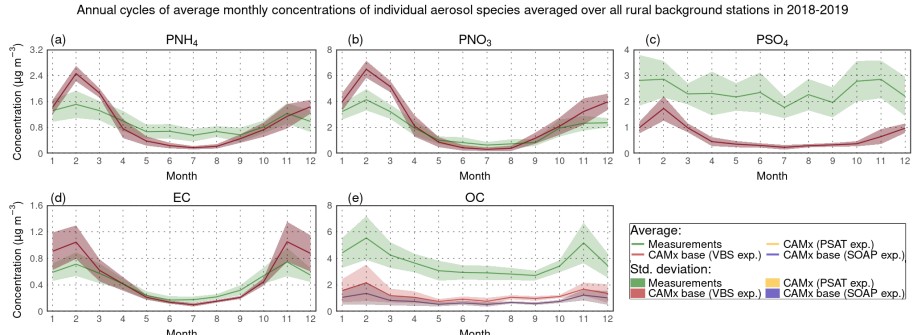

**Figure 3.** Comparison of modeled (the base simulation of the SOAP/VBS experiment – blue/red lines, the simulation of the PSAT experiment – orange lines) and measured (EMEP and AirBase data – green lines) annual cycles of average monthly concentrations of $PNH_4$ **(a)**, $PNO_3$ **(b)**, $PSO_4$ **(c)**, EC **(d)**, and OC **(e)** averaged over all rural background stations during 2018–2019. All concentrations are expressed in $\mu g\ m^{-3}$. The colored areas indicate the standard deviations of the averages, calculated using Eq. (S4) provided in the Supplement. Their color scale corresponds to the scale used for the averages.

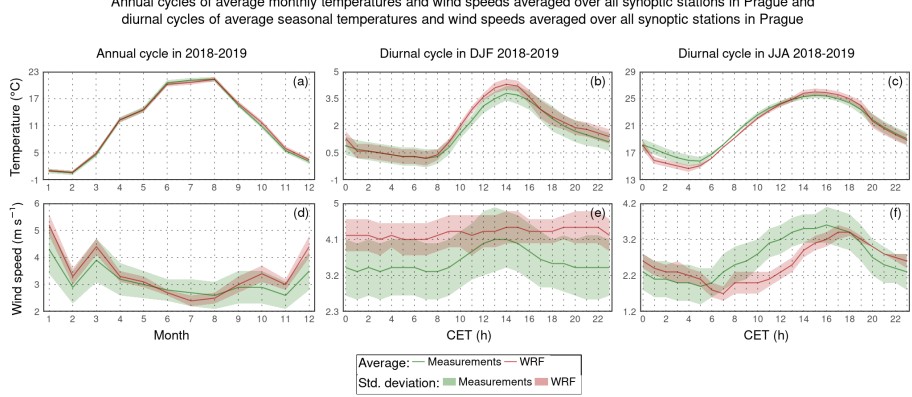

**Figure 4.** Comparison of average modeled (the WRF model – red lines) and measured (CHMI data – green lines) annual cycles of average monthly air temperatures **(a)** and wind speeds **(d)** during 2018–2019, as well as average modeled and measured diurnal cycles of average seasonal air temperatures **(b, c)** and wind speeds **(e, f)** during the winter **(b, e)** and summer **(c, f)** seasons of 2018–2019, where averaging was performed over all Prague synoptic stations. While air temperature is expressed in $^{\circ}$C, wind speed is depicted in $\mathrm{m\,s^{-1}}$. The colored areas indicate the standard deviations of the averages, calculated using Eq. (S4) provided in the Supplement. Their color scale corresponds to the scale used for the averages.

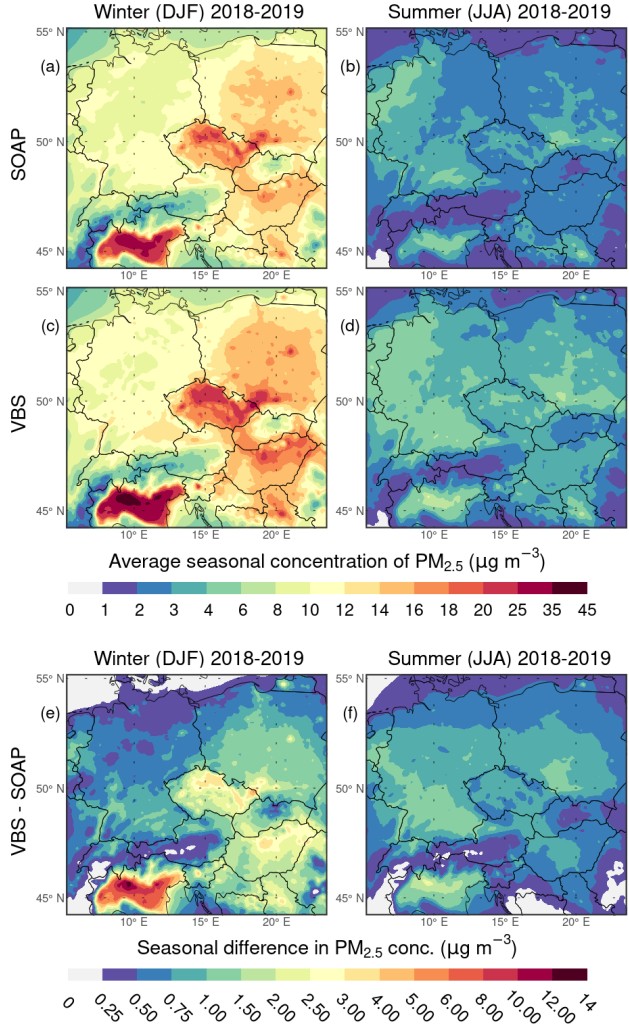

**Figure 5.** Comparison of the average seasonal concentrations of PM$_{2.5}$ (in µg m$^{-3}$) in the base simulations of the SOAP **(a, b)** and VBS **(c, d)** experiments during the winter **(a, c)** and summer **(b, d)** seasons of 2018–2019. Panels **(e)** and **(f)** show the differences between the seasonal PM$_{2.5}$ concentrations in the base simulation of the VBS and SOAP experiments during the winter and summer seasons, respectively.

Average seasonal absolute impact on PM$_{2.5}$ concentration in the SOAP experiment

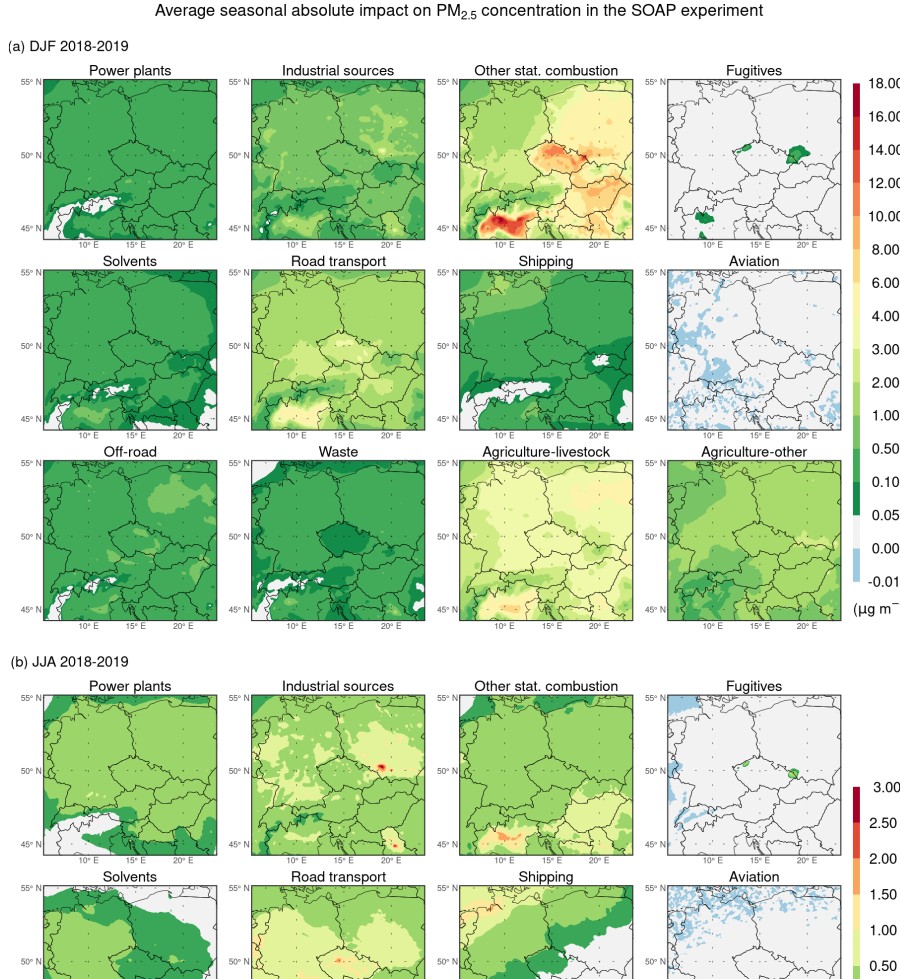

**Figure 6.** Spatial distributions of the average seasonal absolute impact of emissions from individual GNFR sectors A–L (indicated by the sector names in the titles of the subpanels) on the concentration of PM$_{2.5}$ (in µg m$^{-3}$) during the winter **(a)** and summer **(b)** seasons of 2018–2019 in the SOAP experiment.

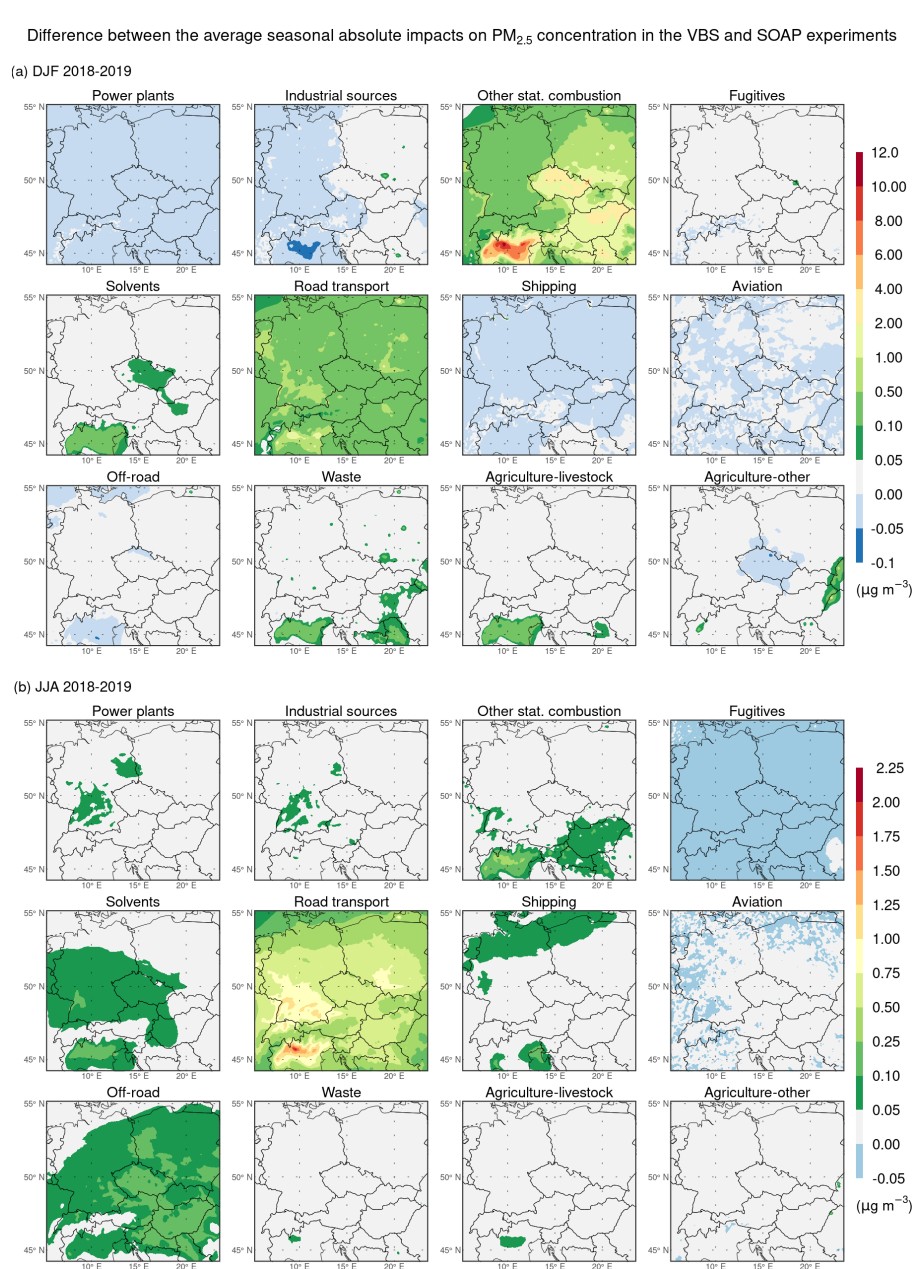

**Figure 7.** Spatial distributions of the differences between the average seasonal absolute impacts of emissions from individual GNFR sectors A–L (indicated by the sector names in the titles of the subpanels) on the concentration of PM$_{2.5}$ (in µg m$^{-3}$) in the VBS and SOAP experiments during the winter **(a)** and summer **(b)** seasons of 2018–2019.

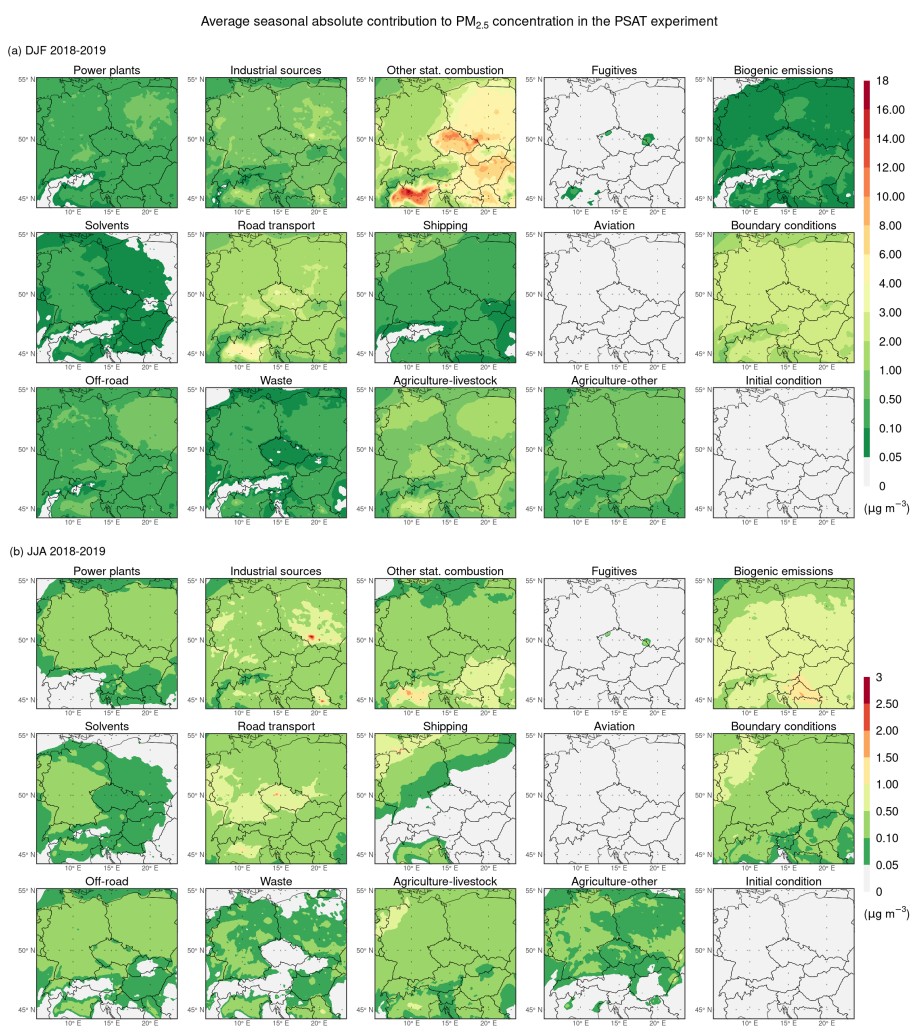

**Figure 8.** Spatial distributions of the average seasonal absolute contribution of emissions from individual categories (indicated in the titles of the subpanels) to the concentration of PM$_{2.5}$ (in µg m$^{-3}$) during the winter **(a)** and summer **(b)** seasons of 2018–2019 in the PSAT experiment. Categories used: GNFR sectors A–L (labeled by the sector names), biogenic emissions, boundary conditions, and initial condition.

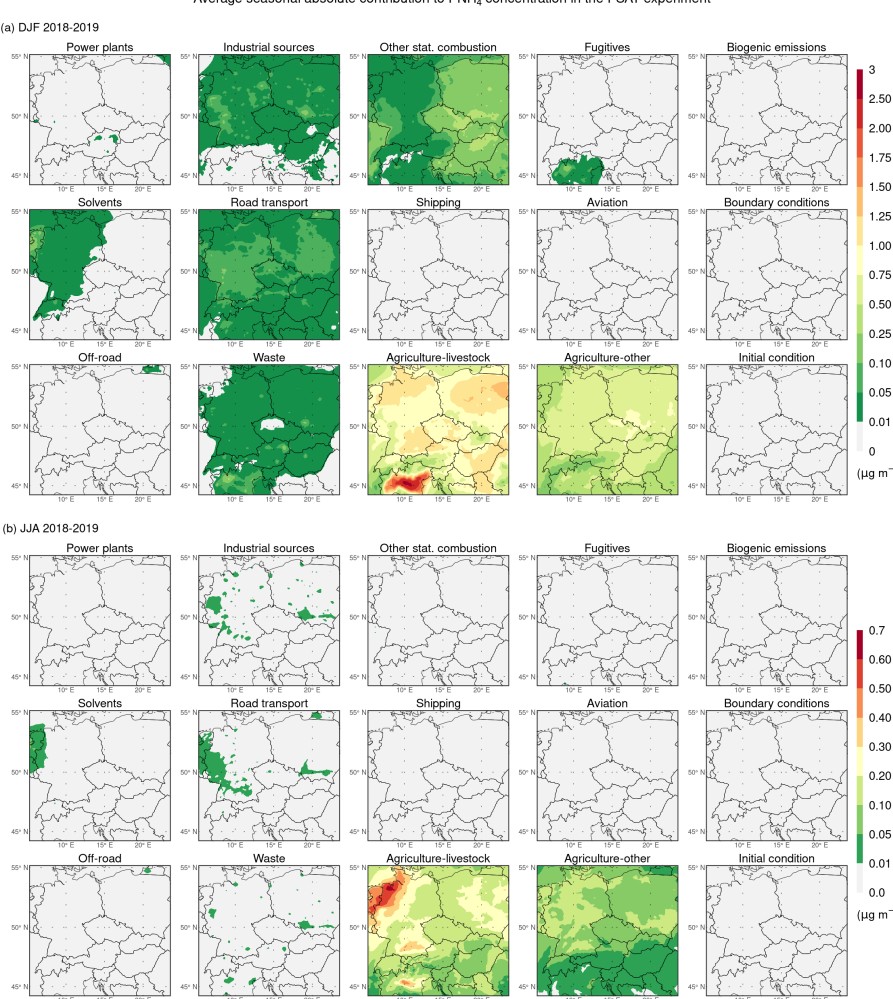

Average seasonal absolute contribution to PNH$_4$ concentration in the PSAT experiment

**Figure 9.** Same as Fig. 8 but for PNH$_4$.

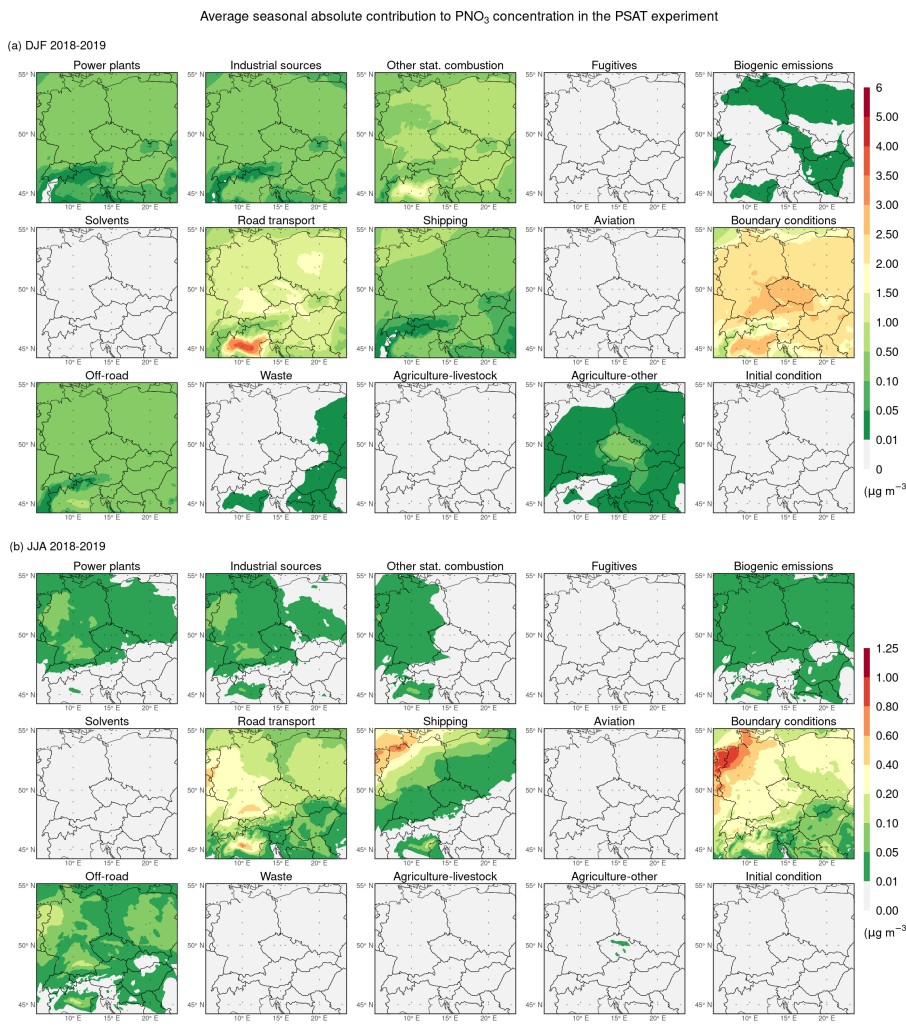

**Figure 10.** Same as Fig. 8 but for PNO₃.

Average seasonal absolute contribution to PSO$_4$ concentration in the PSAT experiment

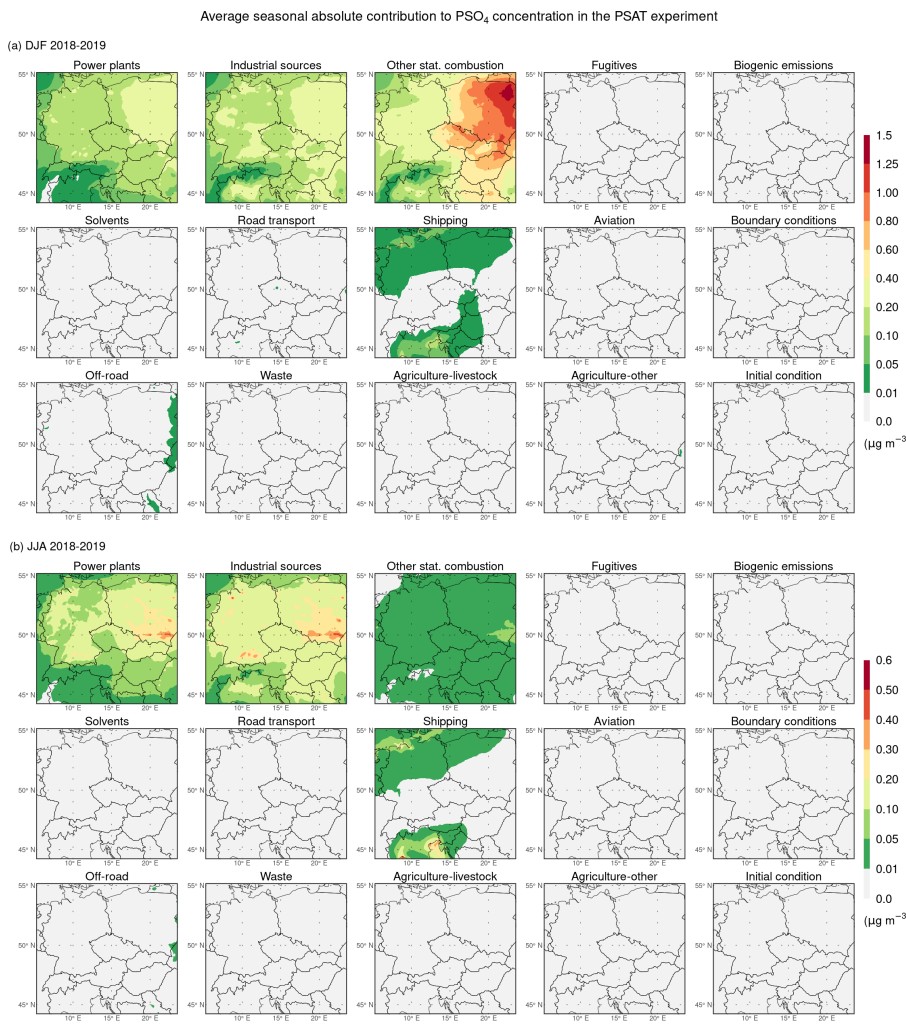

**Figure 11.** Same as Fig. 8 but for PSO$_4$.

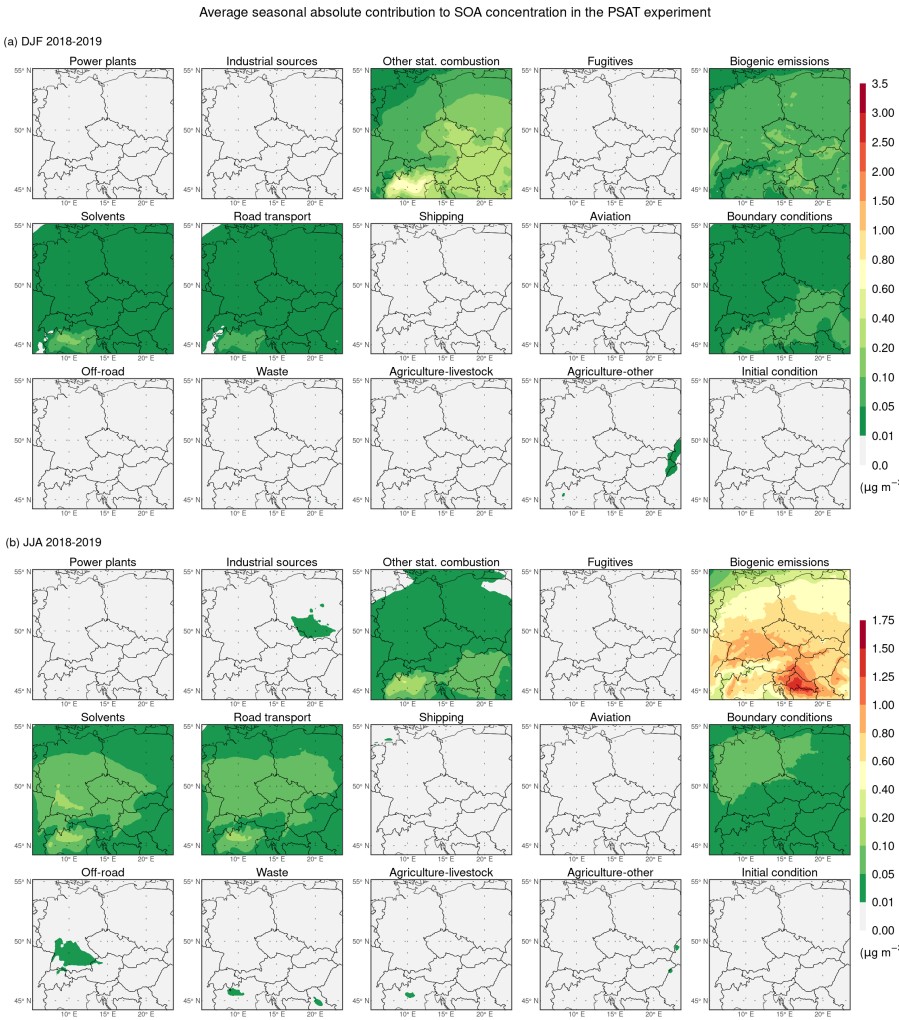

**Figure 12.** Same as Fig. 8 but for SOA.

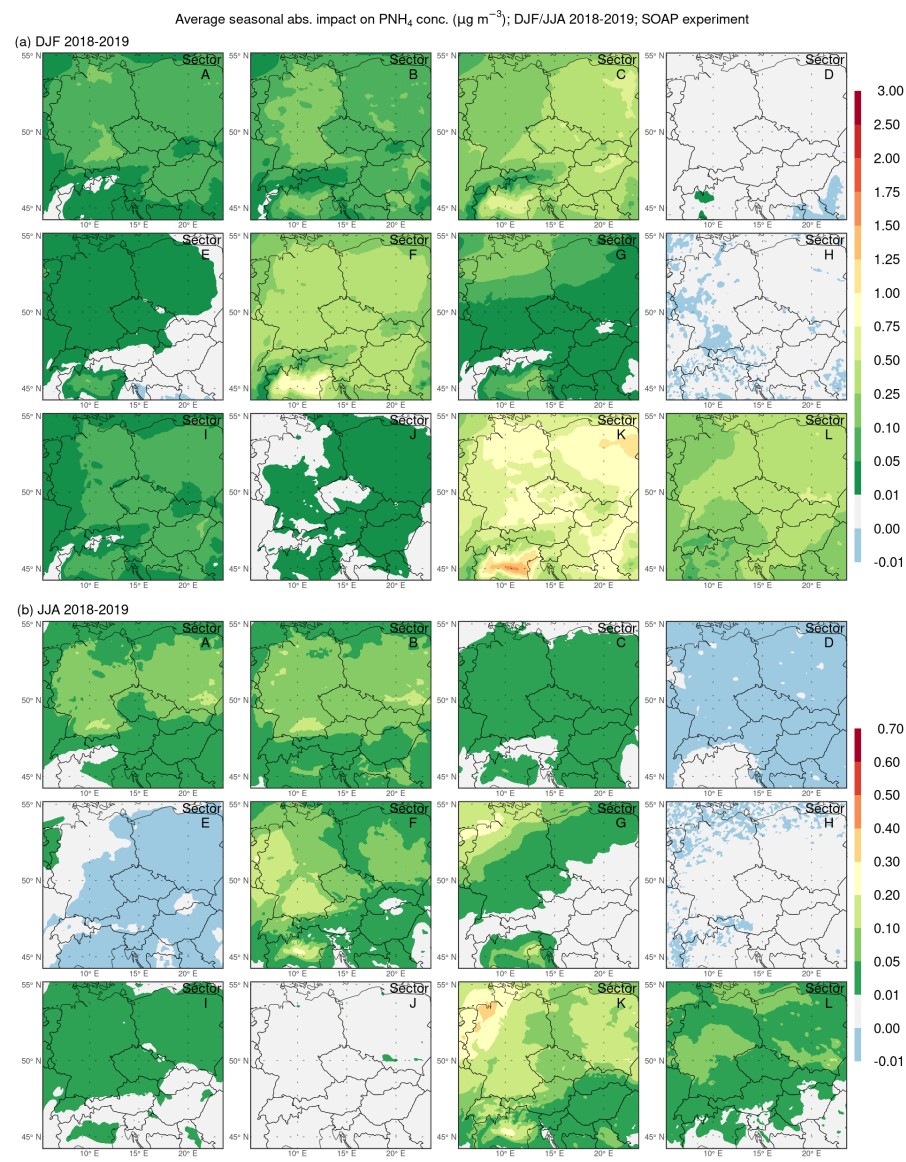

**Figure 13.** Spatial distributions of the average seasonal absolute impact of emissions from individual GNFR sectors A–L (indicated by the sector names in the titles of the subpanels) on PNH$_4$ concentration (in $\mu g\,m^{-3}$) during the winter **(a)** and summer **(b)** seasons of 2018–2019 in the SOAP experiment.

Average seasonal abs. impact on PNO$_3$ conc. (µg m$^{-3}$); DJF/JJA 2018-2019; SOAP experiment

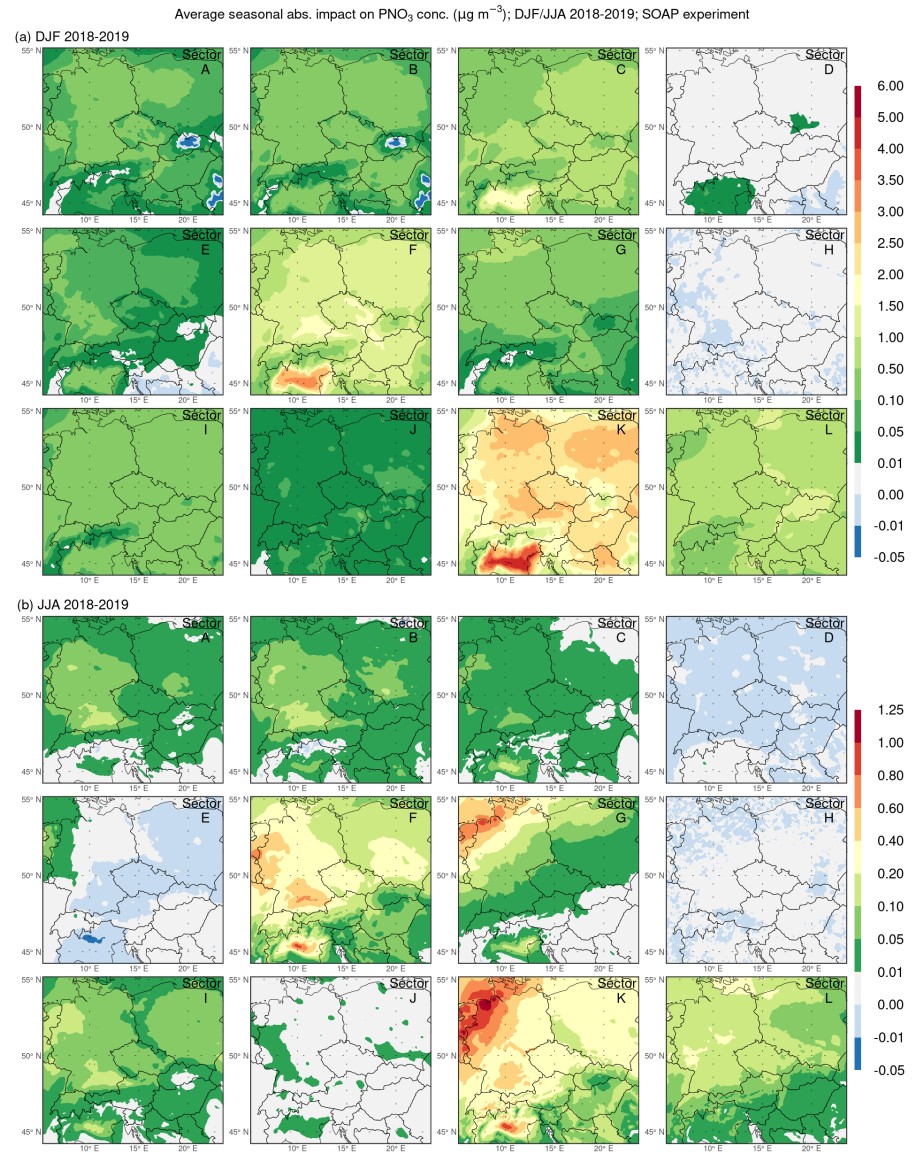

**Figure 14.** Same as Fig. 13 but for PNO$_3$.

Average seasonal abs. impact on $PSO_4$ conc. ($\mu g\ m^{-3}$); DJF/JJA 2018-2019; SOAP experiment

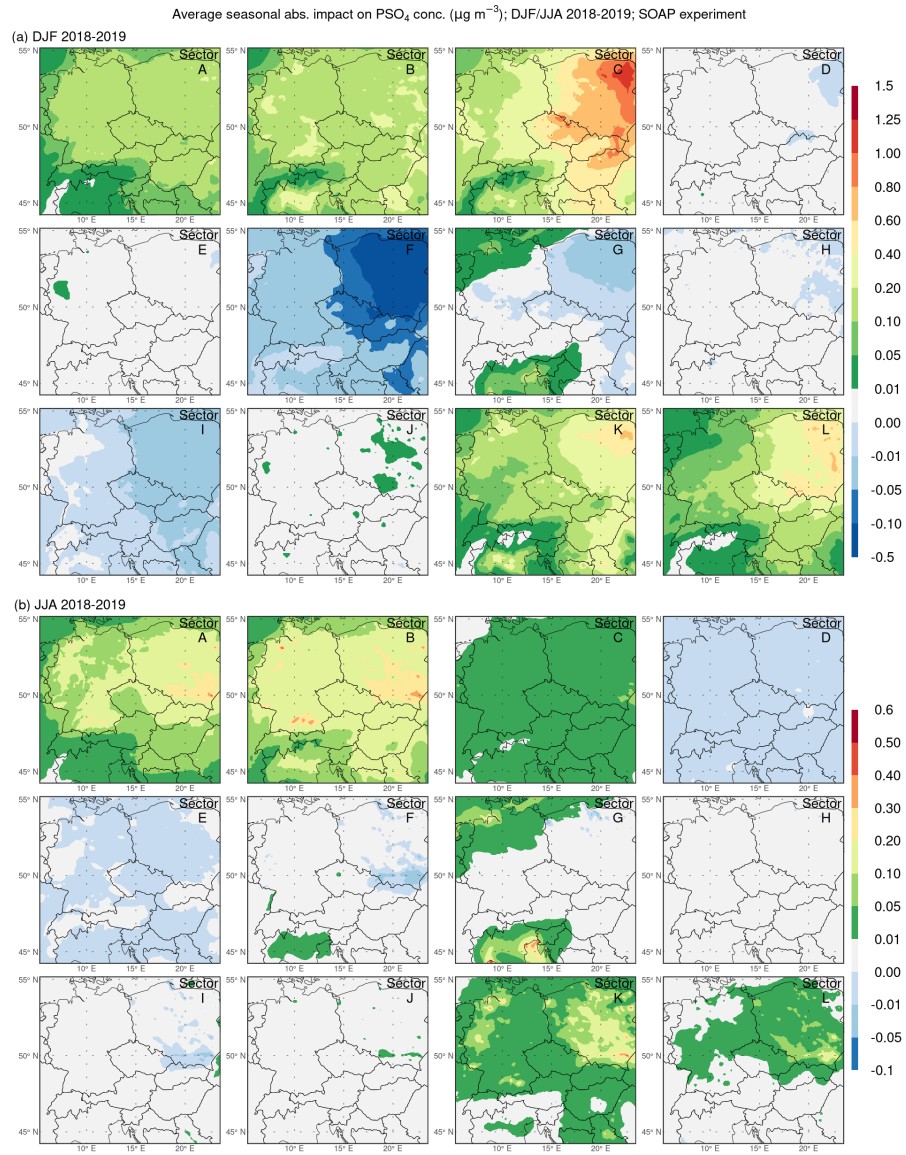

**Figure 15.** Same as Fig. 13 but for $PSO_4$.

Average seasonal abs. impact on SOA conc. (μg m$^{-3}$); DJF/JJA 2018-2019; SOAP experiment

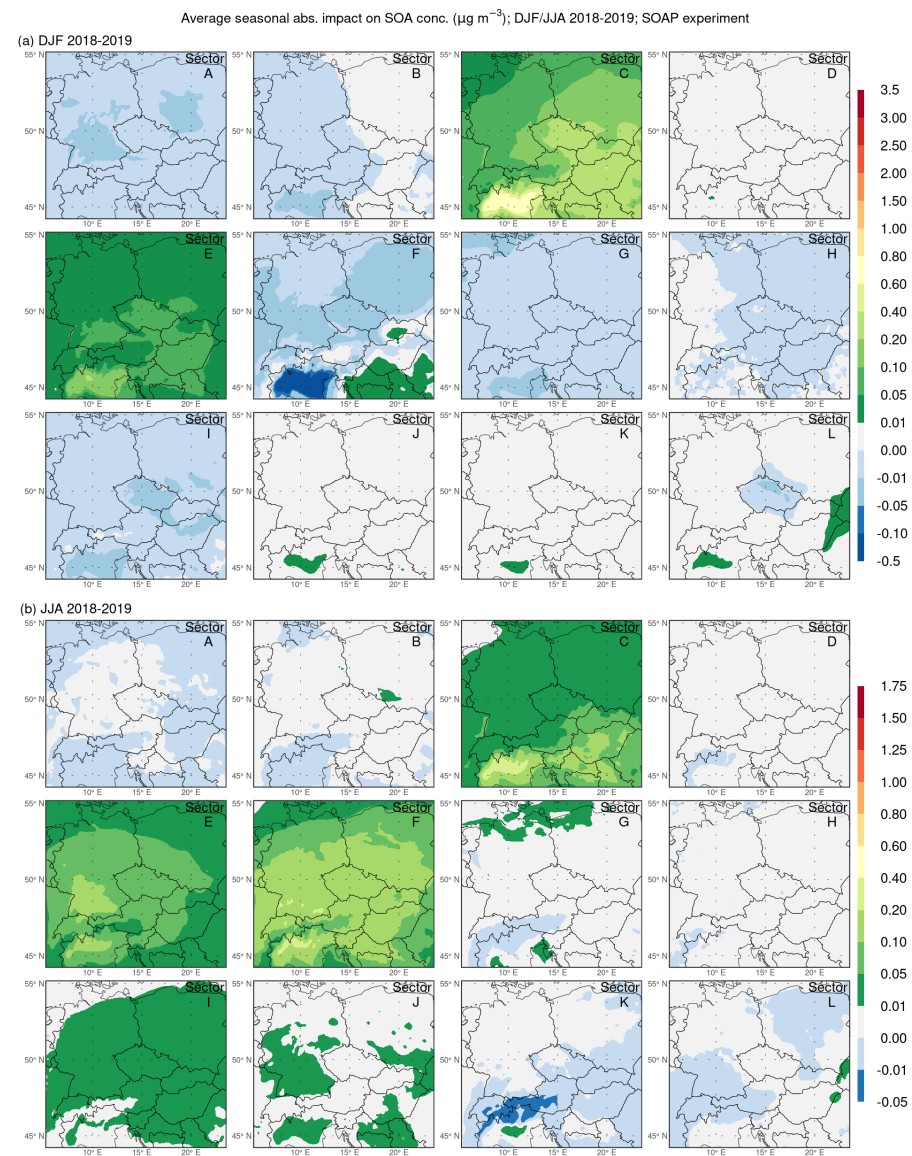

**Figure 16.** Same as Fig. 13 but for SOA.

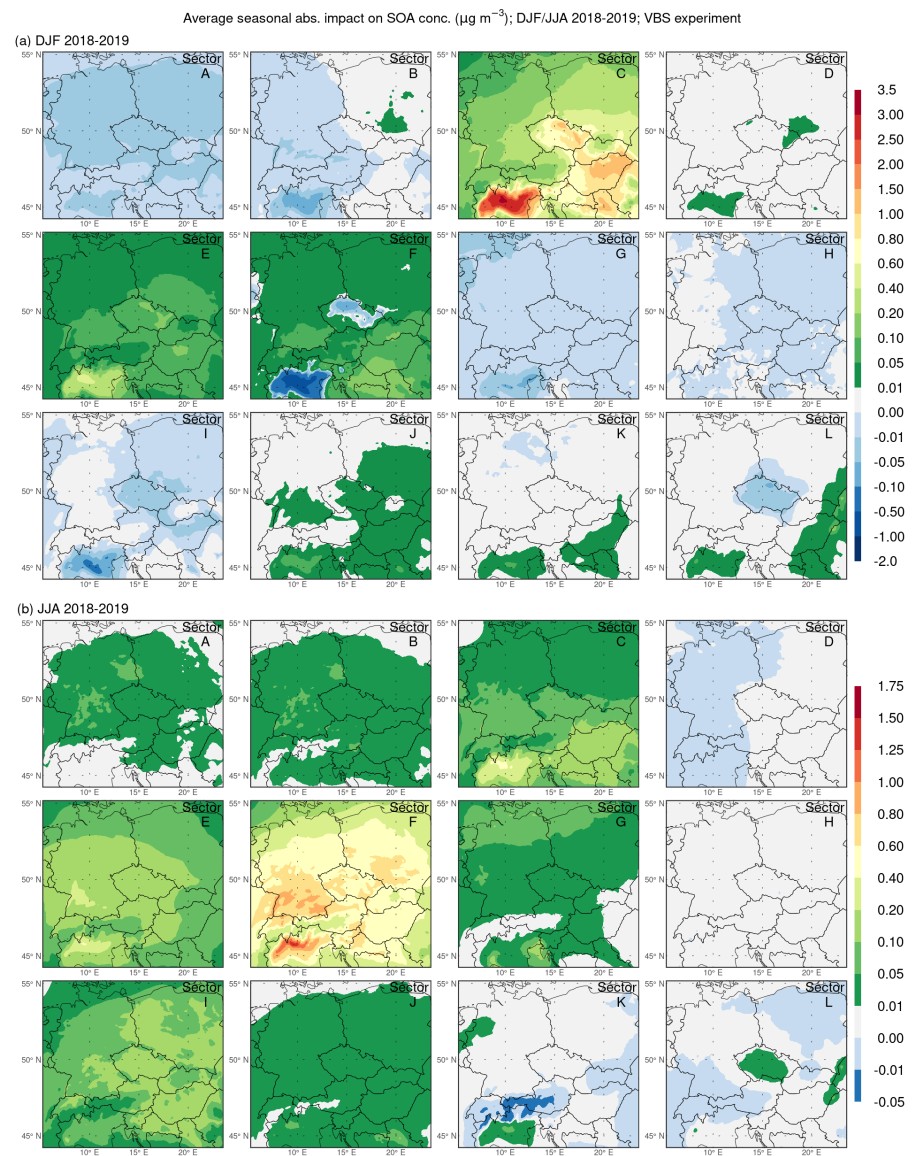

**Figure 17.** Same as Fig. 16 but for the VBS experiment.

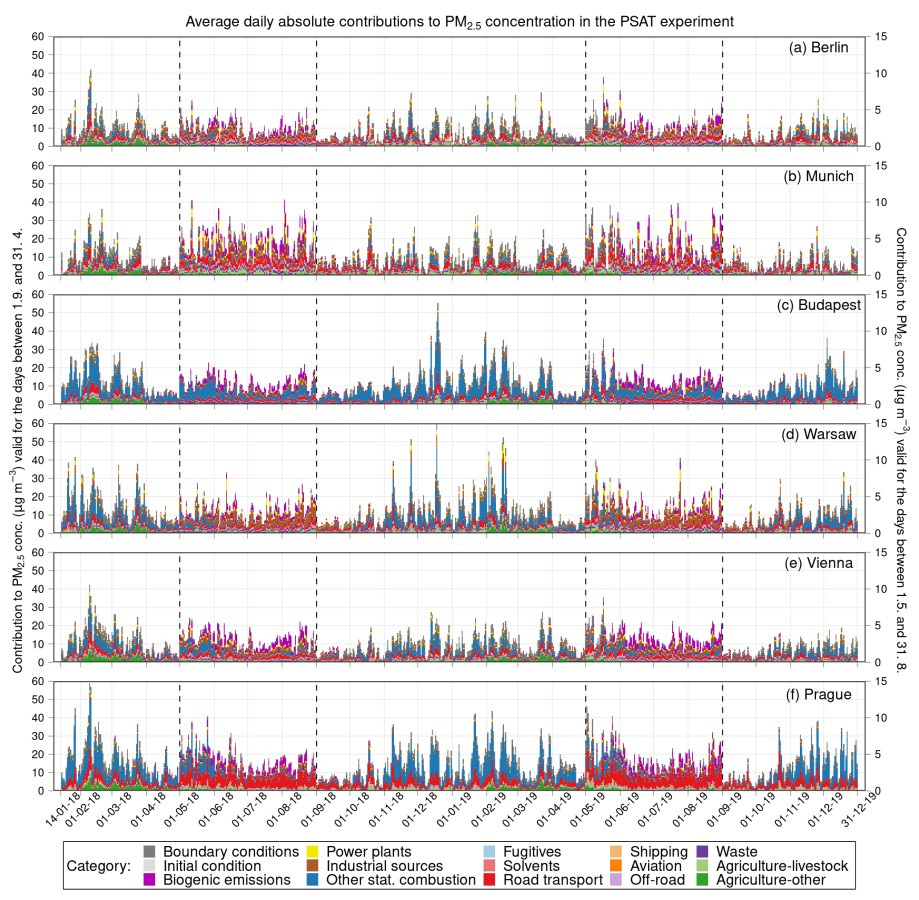

**Figure 18.** Temporal evolution of the average daily absolute contributions of emissions from individual categories to the concentration of PM$_{2.5}$ (in µg m$^{-3}$) above Berlin **(a)**, Munich **(b)**, Budapest **(c)**, Warsaw **(d)**, Vienna **(e)**, and Prague **(f)** in the PSAT experiment. Categories used: GNFR sectors A–L (labeled by the sector names), biogenic emissions, boundary conditions, and initial condition. While the scale on the left side is valid for the days from 1.9. to 31.4., the scale on the right side applies to the days from 1.5. to 31.8.

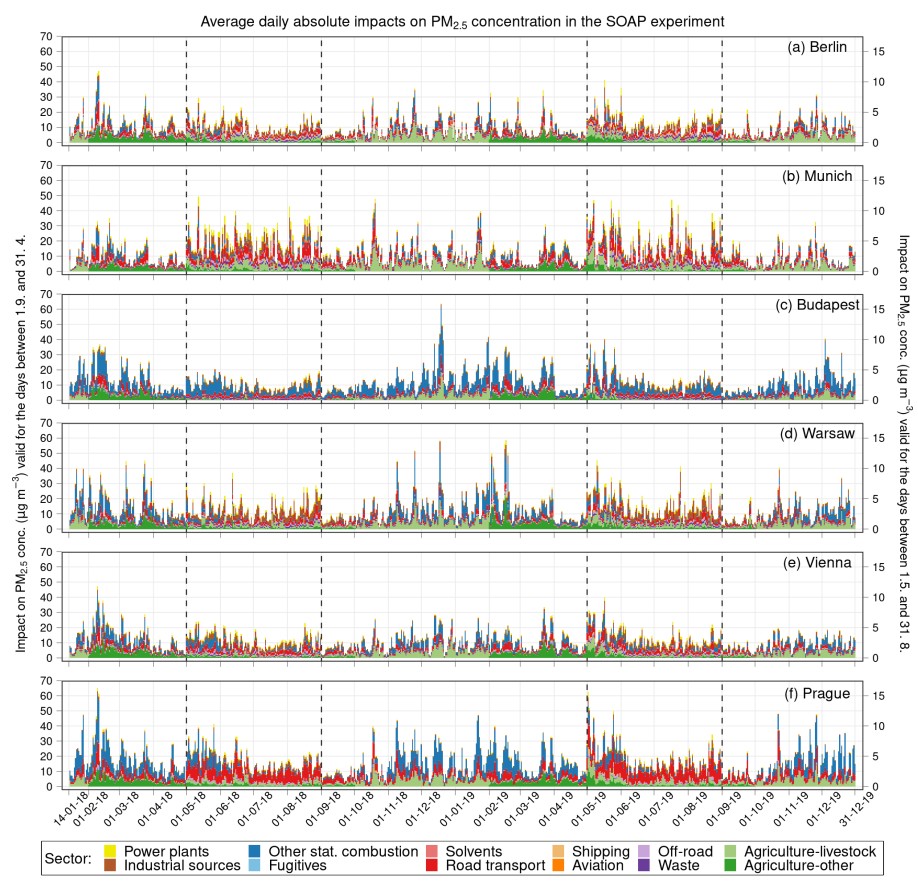

**Figure 19.** Temporal evolution of the average daily absolute impacts of emissions from individual GNFR sectors A–L (labeled by the sector names) on the concentration of $PM_{2.5}$ (in $\mu g\,m^{-3}$) above Berlin **(a)**, Munich **(b)**, Budapest **(c)**, Warsaw **(d)**, Vienna **(e)**, and Prague **(f)** in the SOAP experiment. While the scale on the left side is valid for the days from 1.9. to 31.4., the scale on the right side applies to the days from 1.5. to 31.8.

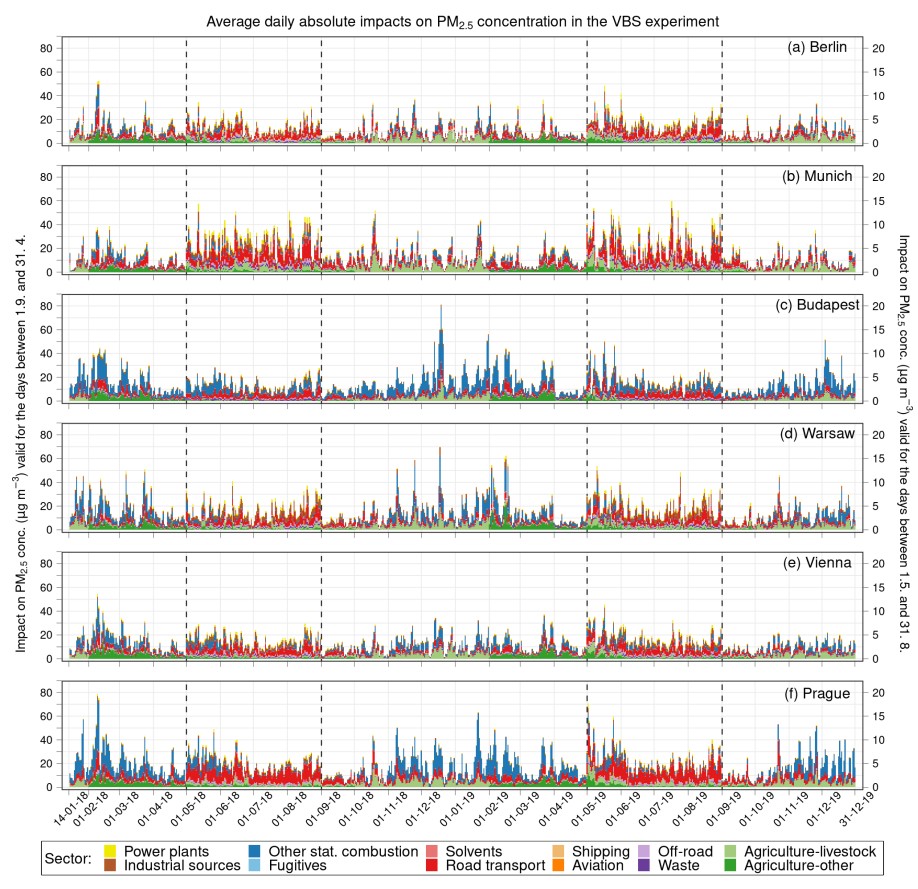

**Figure 20.** Same as Fig. 19 but for the VBS experiment.

**Table 1.** Overview of GNFR sectors used in the study.

| Sector | Sector name |
|--------|-------------|
| A | Power plants |
| B | Industrial sources |
| C | Other stationary combustion |
| D | Fugitives |
| E | Solvents |
| F | Road transport |
| G | Shipping |
| H | Aviation |
| I | Off-road |
| J | Waste |
| K | Agriculture–livestock |
| L | Agriculture–other |

**Table 2.** List of model experiments performed.

| Experiment | Number of simulations | Gas phase mechanism | Inorganic aerosol module | Organic aerosol module | PSAT applied | IVOCs emissions applied | SVOCs emissions applied |
|------------|----------------------|---------------------|--------------------------|------------------------|--------------|-------------------------|-------------------------|
| SOAP | 13* | CB6r5 | ISORROPIA | SOAP | No | Yes | No |
| VBS | 13* | CB6r5 | ISORROPIA | 1.5-D VBS | No | Yes | Yes |
| PSAT | 1 | CB6r5 | ISORROPIA | SOAP | Yes | Yes | No |

* One base and 12 perturbed simulations.

**Table 3.** Comparison of modeled (the base simulation of the SOAP/VBS experiment and the simulation of the PSAT experiment) and measured (AirBase data) daily concentrations of $PM_{2.5}$ in 2018–2019 at suburban and urban stations in Berlin, Munich, Budapest, Prague, Vienna, and Warsaw: evaluation of the Pearson correlation coefficient ($r$), normalized mean bias (NMB, in %), and normalized mean square error (NMSE, in %) averaged over all stations in each city. DJF, MAM, JJA, and SON refer to the winter (December–January–February), spring (March–April–May), summer (June–July–August), and autumn (September–October–November) seasons, respectively.

| City | $PM_{2.5}$ | $r$ | | | NMB (%) | | | NMSE (%) | | |
|---|---|---|---|---|---|---|---|---|---|---|
| | | SOAP | VBS | PSAT | SOAP | VBS | PSAT | SOAP | VBS | PSAT |
| Berlin | DJF | 0.81 | 0.82 | 0.81 | -32.3 | -26.1 | -32.3 | 45.4 | 33.9 | 45.3 |
| | MAM | 0.56 | 0.59 | 0.56 | -51.4 | -45.6 | -51.1 | 102.7 | 79.8 | 101.6 |
| | JJA | 0.55 | 0.62 | 0.55 | -75.8 | -68.0 | -75.6 | 274.3 | 169.42 | 270.7 |
| | SON | 0.70 | 0.73 | 0.71 | -48.6 | -41.0 | -48.4 | 84.0 | 59.6 | 83.2 |
| Munich | DJF | 0.79 | 0.80 | 0.79 | 1.5 | 13.7 | 1.6 | 22.6 | 21.9 | 22.6 |
| | MAM | 0.73 | 0.73 | 0.73 | -33.1 | -23.4 | -32.8 | 66.4 | 48.6 | 65.7 |
| | JJA | 0.48 | 0.51 | 0.48 | -48.2 | -35.1 | -47.6 | 70.4 | 39.5 | 68.4 |
| | SON | 0.72 | 0.73 | 0.72 | -14.1 | -0.4 | -13.7 | 23.7 | 19.2 | 23.4 |
| Budapest | DJF | 0.66 | 0.68 | 0.66 | -19.5 | -3.0 | -19.5 | 32.0 | 23.1 | 31.9 |
| | MAM | 0.43 | 0.45 | 0.43 | -40.8 | -30.7 | -40.5 | 68.2 | 54.3 | 67.4 |
| | JJA | 0.38 | 0.36 | 0.39 | -72.1 | -63.9 | -71.9 | 219.4 | 139.1 | 217.1 |
| | SON | 0.63 | 0.63 | 0.63 | -50.7 | -40.3 | -50.4 | 88.4 | 56.0 | 87.3 |
| Prague | DJF | 0.74 | 0.75 | 0.74 | 5.4 | 19.7 | 5.4 | 28.0 | 28.5 | 28.0 |
| | MAM | 0.58 | 0.59 | 0.58 | -27.5 | -19.6 | -27.2 | 44.9 | 37.4 | 44.4 |
| | JJA | 0.38 | 0.41 | 0.38 | -64.3 | -57.3 | -64.0 | 147.7 | 102.9 | 145.1 |
| | SON | 0.51 | 0.51 | 0.51 | -26.0 | -18.3 | -25.7 | 59.8 | 53.0 | 59.4 |
| Vienna | DJF | 0.65 | 0.67 | 0.65 | -29.4 | -22.2 | -29.4 | 49.7 | 38.0 | 49.6 |
| | MAM | 0.73 | 0.74 | 0.74 | -43.3 | -36.1 | -42.9 | 73.3 | 53.9 | 72.7 |
| | JJA | 0.33 | 0.28 | 0.33 | -63.4 | -52.9 | -63.0 | 138.6 | 84.0 | 135.6 |
| | SON | 0.55 | 0.56 | 0.56 | -49.2 | -41.6 | -48.9 | 117.4 | 88.8 | 115.9 |
| Warsaw | DJF | 0.71 | 0.73 | 0.71 | -21.5 | -9.8 | -21.4 | 30.2 | 22.9 | 30.2 |
| | MAM | 0.68 | 0.70 | 0.68 | -45.9 | -38.3 | -45.7 | 70.2 | 50.7 | 69.7 |
| | JJA | 0.21 | 0.24 | 0.21 | -70.1 | -62.1 | -69.9 | 213.5 | 140.7 | 211.4 |
| | SON | 0.70 | 0.71 | 0.70 | -47.4 | -38.2 | -47.2 | 85.7 | 58.0 | 84.9 |