# Peer review of "Modeling the drivers of fine PM pollution over Central Europe: impacts and contributions of emissions from different sources"

_EGUsphere, 2023_

## Referee Comment (RC2)

**Comments for Modeling the drivers of fine PM pollution over Central Europe: impacts and contributions of emissions from different sources by Bartík et al. (2023)**

Bartík et al. quantified the contributions of various emission sources on the concentration of particulate matter across Central Europe. By applying the PSAT tool and using zero-out method in WRF CAMx model, they determined different major emission sectors for the PM2.5 concentrations for different reasons. They also performed two different group of simulations, SOAP and VBS experiments, based on different organic oxidation chemistry modules. The results of this research can offer recommendations for decreasing air pollution in this region. The results are well discussed with previous literature. The content aligns well with the *ATMOS CHEM PHYS*'s scope and I recommend considering publication after major revisions.

Major comments:

- The analysis of the model simulations shows a certain level of repetition. There are 15 figures, from figure 4 to figure 18, to show the impact of different emission sectors on $PM_{2.5}$ and its components from different cases. Some of the figures can be combined, such as the figure 4 and figure 7. For figures on a similar topic, such as figures 10–13, authors can include one in the main text for detailed discussion while placing the others in the supplement, accompanied by brief descriptions in the main text. By doing so, the differences of impacts of different species can be addressed and it is easier for readers to follow.

- There are too many parentheses in the text and sometimes, using parentheses within parentheses, which strongly affected the clearness of the paper. Please consider moving the text outside the parentheses and revise accordingly, especially for the parentheses in the abstracts.

- Why does only figure 2 depict the analysis at the city level? It would be intriguing to include similar line plots illustrating the emission impacts on $PM_{2.5}$ for these cities as well.

Minor comments:

line 7: delete the species details inside the parentheses.

line 10: Delete "an extreme case of the brute-force method". The abstract should focus on the work done in the paper, not the limitations.

line 11: Full name of GNFR.

line 15: move the text out and revise like "concentrations, with domain-wide average..."

line 15: A space should be used to signify the multiplication of units, such as $\mu g\ m^{-3}$. Please review all the units in the main text to ensure they comply with this rule.

line 19-20: It is better to describe the SOAP experiment first, rather than inside the parentheses.

line 55: Consider deleting the sentences inside the parentheses, as this information is already documented in line 51.

line 88: There is no description of what PSAT is about and what is the difference between PSAT and zero-out method. Since there is a separate group of simulation using PSAT, it is better to add several sentences to document how PSAT calculate the contributions of emission sections.

line 103: Avoid using parentheses within parentheses.

line 104: change "for one winter month (February) and one summer month (August) in 2010" to "during February and August of 2010."

line 115: Consider deleting "(namely for PM2.5, PM10, and coarse PM) "

line 144: Delete '(a detailed description of this revisions is presented in Ramboll (2022))". Repetitive information.

line 147: What is CF?

line 158: Delete "(directly emitted) ".

line 160: Delete "(condensable) ".

line 180: Which wet deposition method in the Seinfeld and Pandis (1998) book? please be detailed.

line 207: Delete "($PM_{2.5}$ and $PM_{10}$) ".

line 212: Change "," to "." after Passant (2002).

line 220: Direct use of SOA. Delete "secondary OA".

line 239: Remove "Since POA is..." out of the brackets.

line 247: Describe the SOAP and VBS simulations in two separate sentences.

line 257: Avoid using parentheses within parentheses.

line 264: Delete "Covering".

line 277: Move the words "we also did ..." in the brackets out.

line 307: Change ":" to ".".

line 308: "Their average underestimate in the base simulation of***" is repeatedly used. Consider revising, or deleting. It is also better to move the sentences out of the brackets.

line 326: Either use a separate sentence to describe the maximum results, or delete those values.

line 356: For these spatial map figures, it would be better to change sector names to emission source names, such as using Power Plants for sector A. It would be easy to understand the plots. There should be no punctuation symbols for figure titles. Please correct accordingly.

line 376: I also recommend using the emission source names, rather than sector symbols to explain the figures. It will help remove lots of redundant words.

line 407: The entire paragraph pertains to a figure in the supplement. It is excessively lengthy, and please condense the descriptions.

line 490: Change ":" to "."

line 525: What's (I)VOC?

line 731: Change "$\approx$" to a word, such as "around".

line 733: same as line 731.

line 769: Remove "who applied..." out of the brackets.

line 777: Remove the bracket and reorganize the sentences inside.

line 793: Remove "the average seasonal absolute contributions of".

line 796: Remove "the average seasonal absolute contributions of".

line 801: Add "from " before "road transport".

---

## Author Comment (AC1)

Authors response on the Anonymous Referee #1 review of "Modeling the drivers of fine PM pollution over Central Europe: impacts and contributions of emissions from different sources"

by Lukáš Bartík et al. (acp-2023-1919)

Dear Anonymous Referee #1,

thank you for your time and effort to review our paper and for all your comments. Please find our point-by-point answers (in black) to the points of your revision (in blue) below.

General comments

The paper "Modeling the drivers of fine PM pollution over Central Europe: impacts and contributions of emissions from different sources" presents a detailed and comprehensive comparison of a source apportionment modelling study based on both brute force and tagging methods.

The paper provides a lot of quantitative results reported in terms of maps and tables that help the reader to evaluate the role of the different sources as well as to understand the differences among the methods.

Therefore, the paper fits the scope of ACP. The paper is also well written, with concise and clear statements, and it does not require any substantial review of syntax and language.

The paper could be published considering just a couple of integrations:

- The model performance evaluation could be supported by a few additional analysis (also in terms of reference) that should consider:
  1. PM precursors (e.g. NOX, NO2, SO2,…)
  2. PM chemical composition (EC, OC, NH4, NO3, SO4,…)
  3. Meteorological variables

This would allow to better investigate the reason of CAMx underestimation, particularly during the summer season

We agree with the reviewer that the validation part has room for extension. Therefore, in the revised manuscript, we reflected all three points raised by the reviewer. In the chemical part, we included a comparison of the modeled gas phases precursors ($NO_2$ and $SO_2$) as well as aerosol components such as elemental carbon (EC), organic carbon (OC) and secondary inorganic species: ammonium ($NH_4^+$), nitrates ($NO_3^-$), and sulfates ($SO_4^{2-}$), as well as a new comparison for $PM_{2.5}$ for available central European stations. In the meteorological part we validated the modeled air temperature and wind speed over Prague where hourly measurements are available. Accordingly, in the revised manuscript, we added the relevant information in the methodology section (Section 2.3, 'Model experiments: design, validation, and evaluation'), with extended Tables S1–S4 in the revised Supplement. Further, in the revised manuscript, we extended the original validation in Section 3.1 ('Validation') with comparison of $PM_{2.5}$ also over Warsaw (Table 3, Figure 2) and added new results in Figures 3 and 4, as well as in Figures S1 and S2 in the revised Supplement. The results are discussed in more detail in revised Section 4 ('Discussion and conclusions') with additional possible reasons that explain the model-observation disagreement (new references added).

- The analysis at the receptor shown in table S1-6 could be extended to a few PM compounds both primary and secondary to better highlight which compounds and which processes give rise to corresponding discrepancies between contributions and impacts shown for PM2.5.

Such analysis could represent an interesting complement to all the maps and could maybe allow to remove some maps (for example some maps with relative contributions that are not so informative)

We accept the fact that the extension of analysis depicted in Tables S1–S6 (in the revised Supplement numbered S5–S10) could offer valuable information and support the statements formulated based on the spatial maps over these selected areas. Therefore, in the revised manuscript, we added a new paragraph dedicated to this new analysis at the end of Section 3.4 ('Impacts and contributions in the selected cities').

Accordingly, we decided to show the seasonal (DJF and JJA) averages of the daily mean impacts/contributions of emissions on the concentrations of all PM$_{2.5}$ components over the studied cities. As there are seven such components (three for primary aerosols and four for secondary aerosol), we added seven Tables numbered S11–S17 in the revised Supplement.

At the same time, we admit that the spatial maps depicting the relative contributions of secondary PM$_{2.5}$ components are not so informative. Therefore, we removed them along with the information from the main text in Section 3.3.2 that referred to these supplemental figures.

**Specific comments and Technical corrections**

**P8 – R250 – which are the differences respect to the setup of "SOAP base case" and "PSAT simulations"?**

The only difference in the model setups between these simulations is in the way the emission inputs are supplied to be able to activate the PSAT module. In the base simulation of the SOAP experiment, we supplied the total emissions calculated with the FUME emission preprocessor. In the PSAT experiment, we divided these total emissions files into one file per category. We clarified this fact in the revised manuscript in Section 2.3. Along with this, we modified Table 2, summarizing the conducted model experiments. Specifically, we extended this table with information on the chemistry schemes/modules for gas-phases chemistry as well as for the aerosol chemistry including the information about the calculation of the emissions of IVOCs and SVOCs.

**P10-R301- Figure 2 – standard deviation bars for modelled results are almost not visible, is it correct?**

Yes, these have been double-checked and they are plotted without mistake. The standard deviation is calculated from the average annual cycles across the selected stations, but we admit that the terminology used could be misleading. Therefore, we clarified this in the methodology section (Section 2.3) and we modified the title of Figure 2 in the revised manuscript to make this clear. We were also consistent with this regarding new Figures 3 and 4, as well as in the case of those in the revised Supplement (Figures S1 and S2).

As for the particular values of the standard deviation (SD), from the new Table S3 in the revised Supplement, it is clear that the number of stations is often small (or sometimes even one only) and these stations are close to each other meaning that the differences between them are small resulting in small SD.

**P10-R302 – SOAP base case and PSAT should provide the same result, isn't it?**

In principle, the species concentrations in the SOAP base and PSAT (that also utilizes the SOAP approach) simulations should be the same as the source apportionment calculation does not influence the pollutant concentrations; it only extends the calculation with traces to track the PM fate. However, there might be numerical differences emerging from the emissions input files. In the case of the SOAP experiment, emissions are summed in the emission preprocessor (FUME) and written into the CAMx input emission Fortran binary file. In the PSAT experiment, a separate input emission file is supplied for each sector. As the precision of emission fluxes is different in FUME and CAMx, numerical rounding causes a difference in the total emissions between the SOAP and PSAT experiments, hence the differences in concentrations. However, these are absolutely minor or negligible without any impact on the results.

In the revised manuscript, we stated the fact of equality of a concentration in the base simulation of the SOAP experiment with the sum of all contributions to that concentration in the PSAT experiment in Section 2.3 ('Model experiments: design, validation, and evaluation'). At the same time, we included the note on the effect of numerical rounding on the slight differences between these two simulations in Section 4 ('Discussion and conclusions').

**P10 – Validation – Which are, according to the authors, the main reason of the discrepancies between modelled and observed PM2.5 values, taking place particularly during the summer season? Are they related to meteorology, lacking in emission inventories?**

**Authors provide some discussion in the final section but keeping it rather generic.**

In the revised manuscript we extended the discussion with the analysis of additional potential sources of the model discrepancies of PM$_{2.5}$, its precursors and components. We based our arguments on the additional

validation that has been included in the revised manuscript and is also mentioned above within our responses.

As for the underestimation of average JJA PM$_{2.5}$ concentrations, we showed within the additional analysis that it holds not only for urban stations but also for sub-urban and rural ones (Figure S2 in the revised Supplement). At the same time from the validation of the PM components (Figure 3 in the revised manuscript) it implies that the model during JJA underestimates especially organic aerosol and sulfates which points to missing/underestimated emissions. Regarding the role of meteorological drivers, we compared our results to a recently published paper, Huszar et al.(2024), who presented a validation of modeled PM$_{2.5}$ and its components for a similar domain, resolution, time-period and using the same model setting in CAMx, and the same emission inputs, however with CAMx driven by RegCM regional model (in contrary to us using WRF for this purpose). Our results are qualitatively very similar to their results but quantitatively they are different. This later might be associated with different meteorological drivers used, however, the qualitative similarities imply problems with emissions.

P11-R292-293 – Did authors expect a larger difference in solvents contribution, with respect to SOAP when applying VBS?

As shown in Figure S11 in the revised version of the Supplement, which corresponds to Figure S18 in the original Supplement, the primary organic aerosol (POA) has a dominant influence on the DJF concentrations of organic aerosol when switching from the SOAP to the VBS scheme. Since POA emissions from solvents are zero (what we know from their annual total), only their NMVOC emissions will contribute to the DJF concentrations of SOA. However, because the DJF concentrations of SOA are much smaller than the DJF concentrations of POA, we did not expect that solvent emissions would have a significant impact after switching the schemes.

P13-R393 – Did author also perform a simulation where "all remaining sources "(and boundary conditions, maybe) are removed? This would allow to check if the sum of all impacts is equal or not to the total concentration of the base case (probably not...)

As our objective was to demonstrate the differences between the two approaches to calculate the effect of anthropogenic emissions, we focused on anthropogenic sectors and the experiments proposed by the reviewer were not conducted. However, we can expect that – as the reviewer states – the sum of all impacts (i.e. also that from biogenic emissions, initial and boundary conditions) will be – due to non-linearity – different from the total concentration in the base case.

P13-R426 – This was expected because impacts and contributions are identical for primary non-reactive compounds.

Yes, this is true. The differences indeed emerge from the secondary aerosol (SA). In the revised manuscript, we noted that no differences are expected in the case of primary non-reactive components.

P15 -R492-496 – from Figure 12 captions and title it seems that maps show the relative fraction with respect to total PNH4 and not to total PM2.5, where the latter seems more reasonable, looking at the maps

Yes, you are correct. The relative contributions of the individual secondary components were indeed calculated with respect to the total PM$_{2.5}$ concentrations (how it is also defined by Eq. A4 in the Appendix of the original manuscript). We admit that the titles used in the Figures mentioned were misleading; however, as said above, these figures were removed from the revised Supplement.

P22 -R718-721 – This statement is reasonable, but it would require additional analysis for example a comparison of modelled and observed PM chemical composition

In the revised manuscript, we added (as already mentioned above) a new analysis focusing on the model-observational differences of the individual aerosol components (namely ammonium, nitrates, sulfates, elemental carbon and organic carbon) and depicted it in Figure 3. These show that the encountered underestimation of PM$_{2.5}$ is caused mostly by underestimated sulfates and organic carbon.

—-------------------------------------------------------------------------

References mentioned:

Huszar, P., Prieto Perez, A. P., Bartík, L., Karlický, J., and Villalba-Pradas, A.: Impact of urbanization on fine particulate matter concentrations over central Europe, Atmos. Chem. Phys., 24, 397–425, https://doi.org/10.5194/acp-24-397-2024, 2024.

Authors response on the Anonymous Referee #2 review of "Modeling the drivers of fine PM pollution over Central Europe: impacts and contributions of emissions from different sources"

by Lukáš Bartík et al. (acp-2023-1919)

Dear Anonymous Referee #2,

thank you for your time and effort to review our paper and for all your comments. Please find our point-by-point answers (in black) to the points of your revision (in blue) below.

Bartík et al. quantified the contributions of various emission sources on the concentration of particulate matter across Central Europe. By applying the PSAT tool and using zero-out method in WRF CAMx model, they determined different major emission sectors for the $PM_{2.5}$ concentrations for different reasons. They also performed two different group of simulations, SOAP and VBS experiments, based on different organic oxidation chemistry modules. The results of this research can offer recommendations for decreasing air pollution in this region. The results are well discussed with previous literature. The content aligns well with the *ATMOS CHEM PHYS*'s scope and I recommend considering publication after major revisions.

Major comments:

- The analysis of the model simulations shows a certain level of repetition. There are 15 figures, from figure 4 to figure 18, to show the impact of different emission sectors on $PM_{2.5}$ and its components from different cases. Some of the figures can be combined, such as the figure 4 and figure 7. For figures on a similar topic, such as figures 10–13, authors can include one in the main text for detailed discussion while placing the others in the supplement, accompanied by brief descriptions in the main text. By doing so, the differences of impacts of different species can be addressed and it is easier for readers to follow.

  We agree that there is room for reducing the number of figures. For example, as the reviewer suggests, the DJF and JJA figures in the case of Figures 4 and 7, 5 and 8, and 6 and 9 were combined into Figures 6, 7, and 8 in the revised manuscript and the Section 3.3.1 titled "$PM_{2.5}$" was revised accordingly. Regarding other figures, we prefer to leave them inside of the manuscript text rather than in the Supplement. The reason is that one of the main outputs of the study is to show that the results – especially in the case of secondary aerosol (both inorganic and organic) – greatly depend on the method used, i.e., these figures show that the impacts (calculated with the zero-out method; SOAP experiment) can be substantially different from the contributions gained from the source apportionment (PSAT) approach. Moreover, the study also aims to address the uncertainty in model calculations of SOA that could potentially strongly influence the impact of different emissions sectors on SOA – justifying to showing also the impact from the VBS run. Therefore, we prefer not to move the associated figures into the Supplement.

- There are too many parentheses in the text and sometimes, using parentheses within parentheses, which strongly affected the clearness of the paper. Please consider moving the text outside the parentheses and revise accordingly, especially for the parentheses in the abstracts.

  We agree that the manuscript contained too many parentheses that strongly affected the integrity of the individual paragraphs. Therefore, the entire text has been carefully revised, and the number of parentheses has been strongly reduced, e.g., by dropping away some parenthesized numeric information.

- Why does only figure 2 depict the analysis at the city level? It would be intriguing to include similar line plots illustrating the emission impacts on $PM_{2.5}$ for these cities as well.

It is not only Figure 2 that is dedicated to the analysis at the city level. Section 3.4, titled "Impacts and contributions in the selected cities", especially focuses on city-level results, with Figures 19–21 in the original manuscript belonging to this part, as well as Figures S16–S17 and Tables S1–S6 in the original Supplement. In the revised manuscript, these are Figures 18–20; in the revised Supplement, these are Figures S9–S10 and Tables S5–S10. In addition, after considering the recommendation of the other reviewer, we expanded the analysis to include individual components of $PM_{2.5}$, which is presented in Tables S11–S17 in the revised Supplement and briefly mentioned in the revised manuscript.

Minor comments:

line 7: delete the species details inside the parentheses.

The species details have been deleted.

line 10: Delete "an extreme case of the brute-force method". The abstract should focus on the work done in the paper, not the limitations.

The mentioned part of the sentence has been deleted.

line 11: Full name of GNFR.

We have replaced this abbreviation with its full name in the revised abstract.

line 15: move the text out and revise like "concentrations, with domain-wide average..."

We have moved the text outside the parentheses and revised it as proposed. To avoid excessive use of parentheses in the abstract, we proceeded analogously in similar cases when the abstract was revised.

line 15: A space should be used to signify the multiplication of units, such as µg m$^{-3}$. Please review all the units in the main text to ensure they comply with this rule.

We have checked all the units used in the manuscript, and if necessary, we edited them so that they met the mentioned rule.

line 19-20: It is better to describe the SOAP experiment first, rather than inside the parentheses.

We edited the abstract so that we briefly presented all the experiments in a separate section before describing the results.

line 55: Consider deleting the sentences inside the parentheses, as this information is already documented in line 51.

The sentence inside the parentheses has been deleted.

line 88: There is no description of what PSAT is about and what is the difference between PSAT and zero-out method. Since there is a separate group of simulation using PSAT, it is better to add several sentences to document how PSAT calculate the contributions of emission sections.

As part of the revision of this paragraph, we have briefly clarified why PSAT is implemented in CAMx, its essence, and the requirements and potential benefits compared to the zero-out method. In the paragraph about the zero-out method, we added a sentence about the fact that this method becomes relatively impractical when studying a large number of emission sources. The fundamental difference between the zero-out method as a representative of the sensitivity method and the PSAT as a representative of the tagged species method was briefly presented in the paragraph above the one describing the zero-out method itself.

line 103: Avoid using parentheses within parentheses.

We rewrote the sentence to avoid the use of parentheses within parentheses.

line 104: change "for one winter month (February) and one summer month (August) in 2010" to "during February and August of 2010."

Changed.

line 115: Consider deleting "(namely for PM$_{2.5}$, PM$_{10}$, and coarse PM) "

Deleted.

line 144: Delete '(a detailed description of this revisions is presented in Ramboll (2022))". Repetitive information.

Deleted.

line 147: What is CF?

The CF scheme designates a static two-mode coarse/fine scheme used in the CAMx model to run aerosol chemistry processes together with the gas-phase chemistry. In the revised manuscript, we have rewritten the sentence in which this abbreviation is used to clarify what it means.

line 158: Delete "(directly emitted) ".

Deleted.

line 160: Delete "(condensable) ".

Deleted.

line 180: Which wet deposition method in the Seinfeld and Pandis (1998) book? please be detailed.

To calculate the wet deposition of gases and aerosols, we applied the CAMx wet deposition model, a detailed description of which can be found in Ramboll (2022). This model employs a scavenging approach in which scavenging coefficients are determined on relationships described by Seinfeld and Pandis (1998). We clarified this in the revised manuscript.

line 207: Delete "(PM$_{2.5}$ and PM$_{10}$) ".

Deleted.

line 212: Change "," to "." after Passant (2002).

Changed.

line 220: Direct use of SOA. Delete "secondary OA".

We deleted 'secondary OA' and used the SOA abbreviation directly instead.

line 239: Remove "Since POA is…" out of the brackets.

Removed out of the parentheses.

line 247: Describe the SOAP and VBS simulations in two separate sentences.

Done.

line 257: Avoid using parentheses within parentheses.

We rewrote the sentence to avoid this.

line 264: Delete "Covering".

Deleted.

line 277: Move the words "we also did …" in the brackets out.

We moved the given sentence outside the parentheses to create a separate sentence.

line 307: Change ":" to ".".

Changed.

line 308: "Their average underestimate in the base simulation of***" is repeatedly used. Consider revising, or deleting. It is also better to move the sentences out of the brackets.

Because the information given in all the parentheses in this paragraph is more or less only supplementary, we decided to omit it in the revised manuscript. The reason for this decision was also the addition of other paragraphs as part of the extension of validation, which was recommended by another reviewer.

line 326: Either use a separate sentence to describe the maximum results, or delete those values.

Again, as the maximum values represent more or less only supplementary information, we decided to omit it in the revised manuscript.

line 356: For these spatial map figures, it would be better to change sector names to emission source names, such as using Power Plants for sector A. It would be easy to understand the plots. There should be no punctuation symbols for figure titles. Please correct accordingly.

We recognize that using sector names instead of emission source names can make it difficult to quickly understand the spatial maps and the time courses of contributions and impacts in the studied cities. That is why we changed them as proposed in all the figures in the main text and the Supplement. When revising the figures, we also considered the note about punctuation symbols in the figure titles and edited them accordingly.

line 376: I also recommend using the emission source names, rather than sector symbols to explain the figures. It will help remove lots of redundant words.

We replaced the sector symbols with the emission source names in the revised manuscript.

line 407: The entire paragraph pertains to a figure in the supplement. It is excessively lengthy, and please condense the descriptions.

In the revised manuscript, we have reduced information about the differences during the winter seasons in the relevant paragraph. However, since we have reorganized Section 3.3.1, we included the corresponding results for the summer seasons, which were previously included in the last paragraph of the section, in this paragraph. Nevertheless, we tried to keep it concise and preserve its main idea.

line 490: Change ":" to ".".

Changed.

line 525: What's (I)VOC?

With this abbreviation, we wanted to express that these are both emissions of VOCs and IVOCs. In the revised manuscript, we have replaced it with 'VOCs and IVOCs emissions' to clarify it.

line 731: Change "≈" to a word, such as "around".

Changed to 'around'.

line 733: same as line 731.

Changed to 'around'.

line 769: Remove "who applied..." out of the brackets.

Removed out of the parentheses.

line 777: Remove the bracket and reorganize the sentences inside.

Moving the sentence out of the parentheses, we made it separate. Next, we reorganized the surrounding sentences.

line 793: Remove "the average seasonal absolute contributions of".

Removed.

line 796: Remove "the average seasonal absolute contributions of".

Removed.

line 801: Add "from " before "road transport".

Added.
* * *
References mentioned:

Ramboll: CAMx User's Guide, Comprehensive Air Quality model with Extentions, version 7.20, Novato, California, available at: http://www.camx.com, 2022.

Seinfeld, J. H. and Pandis, S. N.: Atmospheric Chemistry and Physics: From Air Pollution to Climate Change, Wiley, New York, 1998.

---

## Author Response (AR2)

**Authors' response to the Editor's decision on the revised manuscipt "Modeling the drivers of fine PM pollution over Central Europe: impacts and contributions of emissions from different sources"**

by Lukáš Bartík et al. (acp-2023-1919)

Dear Editor,

thank you for your time and effort to review our revised manuscript and for all your comments. Please find our answer (in black) to your decision (in blue) below.

Thank you for submitting the revised manuscript that has considered the review comments and suggestions. I agree with both referees that the revisions are acceptable. However, the current abstract is too long. According to the ACP manuscript preparation guideline (see https://www.atmospheric-chemistry-and-physics.net/submission.html), the abstract text should be limited to 250 words or less. The referees also offered some suggestions of shortening the abstract. Please revise it accordingly.

We have shortened the abstract considering both the suggestions of both reviewers and the recommendation regarding the limit of the number of words used in it. However, in the case of the recommended limit, we have slightly exceeded it (by about 20 words) at the expense of preserving the integrity and comprehensibility of the new abstract.

In addition, we have corrected several typos that we found in the revised manuscript. These are:

line 202: 'ERA-interim' was changed to 'ERA-Interim'.

line 387: We have changed 'wind speeds wind speeds' to 'wind speeds'.

line 404: Missing unit 'µg m$^{-3}$' was added after the number 45.

line 706: We have changed 'depict' to 'depicts'.

For the sake of completeness, we notify you that we have also added or modified the information in the following sections: Code and data availability, Competing interests, and Acknowledgements. As can be seen, none of these changes, as well as the typo correction changes, affected the scientific content of the revised manuscript.

**Authors' response to the Report #1 written by the Anonymous Referee #2**

Dear Anonymous Referee #2,

thank you for your time and effort to review our revised manuscript and for all your comments. Please find our answer (in black) to your suggestions for a minor revision (in blue) below.

**Suggestions for revision**

Thanks for addressing my comments. I only have minor comments for the revised draft. The current abstract is somewhat verbose. The details about the experiment in line 11-22 seems unnecessary. Additionally, the results parts in the abstract can also be more concise. The contribution sources terms are repetitively mentioned four times between line 23-53, would it be possible to focus on the most important contributor for the emission and concentration in the abstract?

We have shortened the abstract considering both the suggestions you mentioned, i.e.:

- We have removed the unnecessary details about the experiments.

- In the result part, we have focused just on the emission categories that are respondible for the most important contributions and impacts.

Also, we have considered the suggestions of the other referee regarding the shortening of the abstract and the recommendation about the limit of the number of words used in the abstract, which the editor mentioned.

In addition, we have corrected several typos that we found in the revised manuscript. These are:

line 202: 'ERA-interim' was changed to 'ERA-Interim'.

line 387: We have changed 'wind speeds wind speeds' to 'wind speeds'.

line 404: Missing unit '$\mu g\ m^{-3}$' was added after the number 45.

line 706: We have changed 'depict' to 'depicts'.

For the sake of completeness, we notify you that we have also added or modified the information in the following sections: Code and data availability, Competing interests, and Acknowledgements. As can be seen, none of these changes, as well as the typo correction changes, affected the scientific content of the revised manuscript.

**Authors' response to the Report #2 written by the Anonymous Referee #1**

Dear Anonymous Referee #1,

thank you for your time and effort to review our revised manuscript and for all your comments. Please find our answer (in black) to your suggestions for a minor revision (in blue) below.

**Suggestions for revision**

I would consider a minor revision because the abstract is too long. My suggestions are:

- to shorten the first part inclcuding introduction and modelling setup down to 5 rows

- to summarize the obtained results using quantitive information to describe:

    a) the role of the different sources identified by PSAT/contributions and SOAP/impact simulations

    b) the main differences between impacts and contributions

    c) the sensitivity of modelled results between SOAP and VBS

We have shortened the abstract considering all the suggestions you mentioned so that we have either fulfilled them or at least approached them. Also, we have considered the suggestions of the other referee regarding the shortening of the abstract and the recommendation about the limit of the number of words used in the abstract, which the editor mentioned.

In addition, we have corrected several typos that we found in the revised manuscript. These are:

line 202: 'ERA-interim' was changed to 'ERA-Interim'.

line 387: We have changed 'wind speeds wind speeds' to 'wind speeds'.

line 404: Missing unit '$\mu$g m$^{-3}$' was added after the number 45.

line 706: We have changed 'depict' to 'depicts'.

For the sake of completeness, we notify you that we have also added or modified the information in the following sections: Code and data availability, Competing interests, and Acknowledgements. As can be seen, none of these changes, as well as the typo correction changes, affected the scientific content of the revised manuscript.